# How Erdös and Rényi Win the Lottery

## Abstract

Random masks define surprisingly effective sparse neural network models, as has been shown empirically. The resulting Erdös-Rényi (ER) random graphs can often compete with dense architectures and state-of-the-art lottery ticket pruning algorithms struggle to outperform them, even though the random baselines do not rely on computationally expensive pruning-training iterations but can be drawn initially without significant computational overhead. We offer a theoretical explanation of how such ER masks can approximate arbitrary target networks if they are wider by a logarithmic factor in the inverse sparsity $1/\log(1/\text{sparsity})$. While we are the first to show theoretically and experimentally that random ER source networks contain strong lottery tickets, we also prove the existence of weak lottery tickets that require a lower degree of overparametrization than strong lottery tickets. These unusual results are based on the observation that ER masks are well trainable in practice, which we verify in experiments with varied choices of random masks. Some of these data-free choices outperform previously proposed random approaches on standard image classification benchmark datasets.

## 1 Introduction

The impressive breakthroughs achieved by deep learning have largely been attributed to the extensive overparametrization of deep neural networks, as it seems to have multiple benefits for their representational power and optimization (Belkin et al., 2019). The resulting trend towards ever larger models and datasets, however, imposes increasing computational and energy costs that are difficult to meet. This raises the question: Is this high degree of overparameterization truly necessary?

Training general small-scale or sparse deep neural network architectures from scratch remains a challenge for standard initialization schemes Li et al. (2016); Han et al. (2015). However, Frankle & Carbin (2019) have recently demonstrated that there exist sparse architectures that can be trained to solve standard benchmark problems competitively. According to their Lottery Ticket Hypothesis (LTH), dense randomly initialized networks contain subnetworks that can be trained in isolation to a test accuracy that is comparable with the one of the original dense network. Such subnetworks, the lottery tickets (LTs), have since been identified as weak lottery tickets (WLTs) by pruning algorithms that require computationally expensive pruning-retraining iterations (Frankle & Carbin, 2019; Tanaka et al., 2020) or mask learning procedures Savarese et al. (2020); Sreenivasan et al. (2022b). While these can lead to computational gains at training and inference time and reduce memory requirements (Hassibi et al., 1993; Han et al., 2015), the real goal remains to identify good trainable architectures before training, as this could lead to significant computational savings. Yet, contemporary pruning at initialization approaches (Lee et al., 2018; Wang et al., 2020; Tanaka et al., 2020; Fischer & Burkholz, 2022; Frankle et al., 2021) achieve less competitive performance. For that reason it is so remarkable that even iterative state-of-the-art approaches struggle to outperform a simple, computationally cheap, and data independent alternative: random pruning at initialization (Su et al., 2020). Liu et al. (2021) have provided systematic experimental evidence for its 'unreasonable' effectiveness in multiple settings, including complex, large scale architectures and data.

We explain theoretically why they can be effective by proving that randomly masked networks, so called Erdös-Rényi (ER) networks, contain lottery tickets under realistic conditions. Our results imply that sparse ER networks are highly expressive and have the universal function approximation property like dense networks. This insight also provides a missing piece in the theoretical foundation for dynamic sparse training approaches (Evci et al., 2020a; Liu et al., 2021.; Bellec et al., 2018) that start pruning from a random ER network instead of a dense one. The main underlying idea could also

be exploited in different sparsification settings to save computational resources. Up to our knowledge, we are the first to utilize it in the search of strong lottery tickets (SLTs).

Most theoretical results pertaining to LTs focus on the existence of such SLTs (Malach et al., 2020; Pensia et al., 2020; Fischer et al., 2021; da Cunha et al., 2022; Burkholz, 2022a;b). These are subnetworks of large, randomly initialized source networks, which do not require any further training after pruning. Ramanujan et al. (2020) have provided experimental evidence for their existence and suggested that training neural networks could be achieved by pruning alone. By modifying their proposed algorithm, edge-popup (EP), we show experimentally that SLTs are contained in randomly masked ER source networks. We also prove this existence rigorously by transferring the construction of Burkholz (2022a) to random networks. This introduces an additional factor $1/\log(1/\text{sparsity})$ in the lower bound on the width of the source network that guarantees existence with high probability.

In contrast to previous works on SLTs, we also prove the existence of weak LTs. Since every strong LT is also a weak LT, formally, the theory for strong LTs also covers the existence of weak LTs. However, experiments and theoretical derivations suggest that limiting ourselves to SLTs leads to LTs with lower sparsity than what can be achieved by WLT algorithms (Fischer & Burkholz, 2022). In line with this observation, we derive improved existence results for WLTs. However, these cannot overcome the overparameterization factor $1/\log(1/\text{sparsity})$, which is even required under ideal conditions, as we argue by deriving a lower bound on the required width. Our strategy relies on a property of ER networks that is crucial for their effectiveness: They are well trainable with standard initialization approaches. With various experiments on benchmark image data and commonly used neural network architectures, we verify the validity of this assumption, complementing experiments by Liu et al. (2021) for different choices of sparsity ratios. This demonstrates that multiple choices can lead to competitive results. Some of these choices outperform previously proposed random masks Liu et al. (2021), highlighting the potential for tuning sparsity ratios in applications.

**Contributions** In summary, our contributions are as follows: 1) We show theoretically and empirically that ER random networks contain LTs with high probability, if the ER source network is wider than a target by a factor $1/\log(1/\text{sparsity})$. 2) We prove the existence of strong as well as weak LTs in random ER source networks. 3) In support of our theory, we verify in experiments that ER networks are well trainable with standard initialization schemes for various choices of layerwise sparsity ratios. 4) We propose two data-independent, flow preserving and computationally cheap approaches to draw random ER masks defining layerwise sparsity ratios, *balanced* and *pyramidal*. These can outperform previously proposed choices on standard architectures, highlighting the potential benefits resulting from tuning ER sparsity ratios. 5) Our theory explains why ER networks are likely not competitive at extreme sparsities. But this can be remedied by targeted rewiring of random edges as proposed by dynamic sparse training, for which we provide theoretical support with our analysis.

## 1.1 RELATED WORK

Algorithms to prune neural networks can be broadly categorized into two groups, pruning before training and pruning after training. The first group of algorithms that prune after training are effective in speeding up inference, but they still rely on a computationally expensive training procedure (Hassibi et al., 1993; LeCun et al., 1989; Molchanov et al., 2016; Dong et al., 2017; Yu et al., 2022). The second group of algorithms prune at initialization (Lee et al., 2018; Wang et al., 2020; Tanaka et al., 2020; Sreenivasan et al., 2022b) or follow a computationally expensive cycle of pruning and retraining for multiple iterations (Gale et al., 2019; Savarese et al., 2020; You et al., 2019; Frankle & Carbin, 2019; Renda et al., 2019; Dettmers & Zettlemoyer, 2019). These methods find trainable subnetworks, i.e., WLTs (Frankle & Carbin, 2019). Single shot pruning approaches are computationally cheaper but are susceptible to problems like layer collapse which render the pruned network untrainable (Lee et al., 2018; Wang et al., 2020). Tanaka et al. (2020) address this issue by preserving flow in the network through their scoring mechanism. The best performing WLTs are still obtained by expensive iterative pruning methods like Iterative Magnitude Pruning (IMP), Iterative Synflow (Frankle & Carbin, 2019; Fischer & Burkholz, 2022), or training mask parameters of dense networks (Sreenivasan et al., 2022b; Savarese et al., 2020).

However, Su et al. (2020) found that ER masks can outperform expensive iterative pruning strategies in different situations. Inspired by this finding, Golubeva et al. (2021); Chang et al. (2021) have hypothesized that sparse overparameterized networks are more effective than smaller networks with

the same number of parameters. Liu et al. (2021) have further demonstrated the competitiveness of ER masks for two data and pruning free choices of layerwise sparsity ratios across a wide range of neural network architectures and datasets, including complex ones. We show how and when this effectiveness is reasonable. Complementing experiments by Liu et al. (2021), we highlight that ER masks are competitive for various choices of layerwise sparsity ratios.

In addition, we build on the theory for SLTs that prove SLT existence if the widths of the randomly initialized source network (Malach et al., 2020; Pensia et al., 2020; Orseau et al., 2020; Fischer et al., 2021; Burkholz et al., 2022; Burkholz, 2022b; Ferbach et al., 2022) exceeds a value that is proportional to the width of a target network. This theory has been inspired by experimental evidence for SLTs (Ramanujan et al., 2020; Zhou et al., 2019; Diffenderfer & Kailkhura, 2021; Sreenivasan et al., 2022a). The underlying algorithm edge-popup (Ramanujan et al., 2020) finds SLTs by training scores for each parameter of the dense source network and is thus computationally as expensive as dense training. We show that smaller ER masked source networks can be trained instead, as they also contain SLTs, and avoid training a complete network.

However, pruning for SLTs does not seem to achieve as high sparsity ratios as WLT pruning algorithms Fischer & Burkholz (2022), which is also reflected in our existence proofs for SLTs and WLTs. Remarkably, even WLTs require that the ER source network and the resulting LTs are overparameterized relative to the target network, as we show by providing a lower bound on the required width. Our theory suggests that random ER networks face a fundamental limitation at extreme sparsities, as the overparameterization factor scales in this regime as $1/\log(1/(\text{sparsity})) \approx 1/(1 - \text{sparsity})$. This shortcoming could be potentially addressed by targeted rewiring of random edges with Dynamical Sparse Training (DST) that starts pruning from an ER network (Liu et al., 2021.; Evci et al., 2020a;b).

## 2 ERDÖS-RÉNYI NETWORKS AS LOTTERY TICKETS

Our main contribution is to prove the existence of strong and weak lottery tickets in ER networks. In comparison with previous existence results for complete source networks, we require a network width that is larger by a factor $1/\log(1/\text{sparsity})$. To formalize our claims, we first introduce our notation.

**Background, Notation, and Proof-Setup**  Let $\boldsymbol{x} = (x_1, x_2, .., x_d) \in [a_1, b_1]^d$ be a bounded $d$-dimensional input vector, where $a_1, b_1 \in \mathbb{R}$ with $a_1 < b_1$. $f\colon [a_1, b_1]^d \to \mathbb{R}^{n_L}$ is a fully-connected feed forward neural network with architecture $(n_0, n_1, .., n_L)$, i.e., depth $L$ and $n_l$ neurons in Layer $l$. Every layer $l \in \{1, 2, .., L\}$ computes neuron states $\boldsymbol{x}^{(l)} = \phi\left(\boldsymbol{h}^{(l)}\right)$, $\boldsymbol{h}^{(l)} = \boldsymbol{W}^{(l-1)}\boldsymbol{x}^{(l-1)} + \boldsymbol{b}^{(l-1)}$. $\boldsymbol{h}^{(l)}$ is called the pre-activation, $\boldsymbol{W}^{(l)} \in \mathbb{R}^{n_l \times n_{l-1}}$ is the weight matrix and $\boldsymbol{b}^{(l)}$ is the bias vector. We also write $f(\boldsymbol{x}; \theta)$ to emphasize the dependence of the neural network on its parameters $\theta = (\boldsymbol{W}^{(l)}, \boldsymbol{b}^{(l)})_{l=1}^L$. For simplicity, we restrict ourselves to the common ReLU $\phi(x) = \max\{x, 0\}$ activation function, but most of our results can be easily extended to more general activation functions as in (Burkholz, 2022b;a). In addition to fully-connected layers, we also consider convolutional layers. For a convenient notation, without loss of generality, we flatten the weight tensors so that $\boldsymbol{W}_T^{(l)} \in \mathbb{R}^{c_l \times c_{l-1} \times k_l}$ where $c_l, c_{l-1}, k_l$ are the output channels, input channels and filter dimension respectively. For instance, a 2-dimensional convolution on image data would result in $k_l = k'_{1,l}k'_{2,l}$, where $k'_{1,l}, k'_{2,l}$ define the filter size.

We distinguish two kinds of neural networks, a target network $f_T$ and a source network $f_S$. $f_T$ is approximated by a lottery ticket (LT) that is a obtained by masking the parameters of $f_S$ and, in case of a weak LT, learning of the parameters. We assume that $f_T$ has depth $L$ and parameters $\left(\boldsymbol{W}_T^{(l)}, \boldsymbol{b}_T^{(l)}, n_{T,l}, m_{T,l}\right)$ are the weight, bias, number of neurons and number of nonzero parameters of the weight matrix in Layer $l \in \{1, 2, .., L\}$. Note that this implies $m_l \le n_l n_{l-1}$. Similarly, $f_S$ has depth $L + 1$ with parameters $\left(\boldsymbol{W}_S^{(l)}, \boldsymbol{b}_S^{(l)}, n_{S,l}, m_{S,l}\right)_{l=0}^L$. Note that $l$ ranges from 0 to $L$ for the source network, while it only ranges from 1 to $L$ for the target network. The extra source network layer $l = 0$ accounts for an extra layer that we need in our LT construction.

**ER networks**  Instead of a complete source network, we will consider random Erdös-Rényi (ER) networks $f_{ER} \in ER(\mathbf{p})$ with layerwise sparsity ratios $p_l$. $f_{ER}$ can be defined as subnetwork of a complete source network using a binary mask $\boldsymbol{S}_{ER}^{(l)} \in \{0,1\}^{n_l \times n_{l-1}}$ or $\boldsymbol{S}_{ER}^{(l)} \in \{0,1\}^{n_l \times n_{l-1} \times k_l}$ for every layer. The mask entries are drawn from independent Bernoulli distributions with layerwise success probability $p_l > 0$, i.e., $s_{ij,ER}^{(l)} \sim \text{Ber}(p_l)$. The random pruning is performed initially with negligible computational overhead and the mask stays fixed during training. Note that $p_l$ is also the expected density of that layer. The overall expected density of the network is given as $p = \frac{\sum_l m_l p_l}{\sum_k m_k} = 1 - \text{sparsity}$. In case of uniform $p_l = p$, we also write $ER(p)$ instead of $ER(\mathbf{p})$. An ER neural network is defined as $f_{ER} = f_S(\boldsymbol{x}; \boldsymbol{W} \cdot \boldsymbol{S}_{ER})$. We will also call $f_{ER} \in ER(\mathbf{p})$ the source network, as we might need to prune additional parameters during a LT construction. Here $\mathbf{p}$ denotes the vector of layerwise success probabilities. This pruning defines another mask $S_{LT}$, which is a subnetwork of $S_{ER}$, i.e., a zero entry $s_{ij,ER} = 0$ implies also a zero in $s_{ij,LT} = 0$, but the converse is not true. We skip the subscripts LT or ER if the nature of the mask is clear from the context.

## 2.1 Existence of Strong Lottery Tickets

Most strong lottery ticket (SLT) existence proofs construct explicitly a LT that approximates any target network of a given width and depth. The LT is defined by a mask that encodes the result of pruning a randomly initialized source network. Most proofs that derive a logarithmic lower bound on the overparametrization factor (i.e., the factor by which the source network is supposed to be wider than the target network) (Pensia et al., 2020; Burkholz et al., 2022; Burkholz, 2022a; da Cunha et al., 2022; Burkholz, 2022b; Ferbach et al., 2022), solve multiple subset sum approximation problems (Lueker, 1998). For every target parameter $z$, they identify some random parameters of the source network $X_1, ..., X_n$ that can be masked or not to approximate $z$. In case of an ER source network, $1 - p$ random connections are missing in comparison with a dense source network. These missing connections also reduce the amount of available source parameters $X_1, ..., X_n$. To take this into account, we modify the corresponding subset sum approximations according to the following lemma.

**Lemma 2.1** (Subset sum approximation in ER Networks). *Let $X_1, ..., X_n$ be independent, uniformly distributed random variables so that $X_i \sim U([-1, 1])$ and $M_1, ..., M_n$ be independent, Bernoulli distributed random variables so that $M_i \sim \text{Ber}(p)$ for a $p > 0$. Let $\epsilon, \delta \in (0, 1)$ be given. Then for any $z \in [-1, 1]$ there exists a subset $I \subset [n]$ so that with probability at least $1 - \delta$ we have $|z - \sum_{i \in I} M_i X_i| \leq \epsilon$ if*

$$n \geq C \frac{1}{\log(1/(1-p))} \log \left( \frac{1}{\min(\delta, \epsilon)} \right). \tag{1}$$

The proof is given in Appendix A.2 and utilizes the original subset sum approximation result for random subsets of the base set $X_1, ..., X_n$. In addition, it solves the challenge to combine the involved constants respecting the probability distribution of the random subsets. For simplicity, we have formulated it for uniform random variables and target parameters $z \in [-1, 1]$ but it could be easily extended to random variables that contain a uniform distribution (like normal distributions) and generally bounded targets as in Corollary 7 in (Burkholz et al., 2022).

In comparison with the original subset sum approximation result, we need a base set that is larger by a factor $\frac{1}{\log(1/(1-p))}$. This is exactly the factor by which we can modify contemporary SLT existence results to transfer to ER source networks. By replacing the subset sum approximation construction with Lemma 2.1, we can thus show SLT existence for fully-connected (Pensia et al., 2020; Burkholz, 2022b), convolutional (Burkholz et al., 2022; Burkholz, 2022a; da Cunha et al., 2022), and residual ER networks (Burkholz, 2022a) or random GNNs (Ferbach et al., 2022). To give an example for the effective use of this lemma and discuss the general transfer strategy, we explicitly extend the SLT existence results by Burkholz (2022b) for fully-connected networks to ER source networks. We thus show that pruning a random source network of depth $L + 1$ with widths larger than a logarithmic factor can approximate any target network of depth $L$ with a given probability $1 - \delta$.

**Theorem 2.2** (Existence of SLTs in ER Networks). *Let $\epsilon, \delta \in (0, 1)$, a target network $f_T$ of depth $L$, an $ER(\mathbf{p})$ source network $f_S$ of depth $L + 1$ with edge probabilities $p_l$ in each layer $l$ and iid initial parameters $\boldsymbol{\theta}$ with $w_{ij}^{(l)} \sim U([-1, 1]), b_i^{(l)} \sim U([-1, 1])$ be given. Then with probability at least $1 - \delta$, there exists a mask $\boldsymbol{S}_{LT}$ so that each target output component $i$ is approximated as*

$\max_{\boldsymbol{x} \in \mathcal{D}} \|f_{T,i}(\boldsymbol{x}) - f_{S,i}(\boldsymbol{x}; \boldsymbol{W}_S \cdot \boldsymbol{S}_{LT})\| \leq \epsilon$ if

$$n_{S,l} \geq C \frac{n_{T,l}}{\log\left(1/(1 - p_{l+1})\right)} \log\left(\frac{1}{\min\{\epsilon_l, \delta/\rho\}}\right)$$

for $l \geq 1$, where $\epsilon_l = g(\epsilon, f_T)$ is defined in Appendix A.2 and $\rho = \frac{C N_T^{1+\gamma}}{\log(1/(1 - \min_l p_l))^{1+\gamma}} \log(1/\min\{\min_l \epsilon_l, \delta\})$ for any $\gamma \geq 0$. We also require $n_{S,0} \geq Cd \frac{1}{\log(1/(1-p_1))} \log\left(\frac{1}{\min\{\epsilon_1, \delta/\rho\}}\right)$, where $C > 0$ denotes a generic constant that is independent of $n_{T,l}$, $L$, $p_l$, $\delta$, and $\epsilon$.

*Proof Outline*: The main LT construction idea is visualized in Fig. 3 (b). For every target neuron, multiple approximating copies are created in the respective layer of the LT to serve as basis for modified subset sum approximations (see Lemma 2.1) of the parameters that lead to the next layer. In line with this approach, the first layer of the LT consists of univariate blocks that create multiple copies of the input neurons. In addition to Lemma 2.1, also the total number of subset sum approximation problems $\rho$ that have to be solved needs to be re-assessed for ER source networks, as this influences the probability of LT existence. This modification is driven by the same factor as the width increase. The full proof is given in Appendix A.2.

**Experiments for SLTs** To verify the existence of SLTs in ER networks experimentally, we have conducted experiments with a ResNet18 model on CIFAR10. Average results based on 3 independent repetitions are shown in Table 1. We initialize the ResNet18 as a sparse ER network and use the edge-popup (Ramanujan et al., 2020) algorithm to find a SLT, restricting edge-popup only to the nonzero (unmasked) parameters in the ER network as explained in Fig. 1. Our results show that it is possible to obtain SLTs by starting pruning from a sparse random network instead of a dense network. Importantly, we can start with a sparse ER network of up to $0.8$ sparsity and still achieve competetive performance to find a SLT with final sparsity $0.9$, without the need to train a dense network from scratch. Additional experiments for ResNet18 and VGG16 on CIFAR10 and ResNet110 on CIFAR100 are presented in the appendix (see, e.g., Table 13 and 15).

| Sparsity | $0 \to 0.9$ | $0.5 \to 0.9$ | $0.7 \to 0.9$ | $0.5 \to 0.95$ | $0.8 \to 0.95$ | $0.5 \to 0.99$ |
|---|---|---|---|---|---|---|
| Test Acc. | $87.9 \pm 0.2$ | $88.1 \pm 0.3$ | $88.0 \pm 0.3$ | $87.8 \pm 0.3$ | $88.1 \pm 0.1$ | $87.9 \pm 0.1$ |

Table 1: *ER networks for Strong Lottery Tickets*: Average results and $0.95$ standard confidence intervals for training an ER ResNet18 network with edge popup (Ramanujan et al., 2020) on CIFAR10. The ER network is initialized with a uniform initial sparsity, which is gradually annealed to attain a SLT of the final sparsity (initial $\to$ final sparsity). Note that the first column serves as baseline.

## 2.2 EXISTENCE OF WEAK LOTTERY TICKETS

LT existence proofs have been restricted to SLTs, which do not need to be trained after pruning. These proofs (Fischer et al., 2021; Burkholz, 2022b; Pensia et al., 2020) automatically hold for WLTs, as every SLT is also a WLT. Hence, SLTs and WLTs in ER networks exist as we have shown with Theorem 2.2. However, the constructed LTs have usually more parameters than the target network, as the sum over multiple random weight parameters is used to approximate a single target parameter. If we could use only one parameter in the source network for every target we could potentially achieve sparser LTs. The reason that LT theory is primarily focused on SLTs is because it is not well understood what properties and parameter initialization schemes render arbitrarily pruned networks trainable with SGD Li et al. (2016); Han et al. (2015). ER masks, however, do not seem to suffer from this limitation, as they have been found to be well trainable in practice (Su et al., 2020; Ma et al., 2021; Liu et al., 2021), which we also verify in experiments in Section 3 in a broader context with different choices of layerwise sparsity ratios. The following assumption underlying our WLT existence proofs is therefore justified. ER networks are well trainable with SGD for different layerwise sparsity ratios and standard weight initialization schemes.

**Assumption 2.3** (ER networks are trainable). An ER network $f \in ER(\mathbf{p})$ with layerwise density vector $\mathbf{p}$ is trainable by SGD with standard weight initializations (He et al., 2015).

Based on this assumption, we provide a construction of a LT by deriving an exact representation of the target network that only uses existing edges of a random ER network.

## 2.3 WLT Existence for a Single Hidden Layer Target Network

We start with showing WLT existence for a single hidden layer fully-connected target network. Our proof strategy is visually explained in Figure 1. To approximate a target network with a single hidden layer, we use a two hidden layer ER source network. For a given density, the following theorem identifies the width of an ER source network, above which we can show existence.

**Theorem 2.4** (Existence of an ER network as a WLT for a single hidden layer target network). *Let Assumption 2.3 be fulfilled and a single layer fully-connected target network $f_T(\boldsymbol{x}) = \boldsymbol{W}_T^{(2)}\phi(\boldsymbol{W}_T^{(1)}\boldsymbol{x} + \boldsymbol{b}_T^{(1)}) + \boldsymbol{b}_T^{(2)}$, $\delta \in (0,1)$, a target density $p$ and a 2 layer ER source network $f_S \in ER(\mathbf{p})$ with widths $n_{S,0} = q_0 d, n_{S,1} = q_1 n_{T,1}, n_{S,2} = q_2 n_{T,2}$ be given. If*

$$q_0 \geq \frac{1}{\log(1/(1-p_1))}\log\left(\frac{2m_{T,1}q_1}{\delta}\right), q_1 \geq \frac{1}{\log(1/(1-p_2))}\log\left(\frac{2m_{T,2}}{\delta}\right) \text{ and } q_2 = 1,$$

*then with a probability $1 - \delta$ the source network $f_S$ contains a weak LT $f_{WLT}$.*

*Proof Outline*: Similar to the strategy for SLTs, we create an univariate first layer in the source network as explained in Fig. 1. Different from the subset sum approximation in case of SLTs, we can now use the trainability assumption 2.3 to choose a weight in the source ER network which exactly learns the value of a target weight. The key idea is to create multiple copies (blocks in Fig. 1 (b)) in the source network for each target neuron such that every target link is realized by pointing to at least one of these copies in the ER source. In the appendix, we derive the corresponding weight and bias parameters that can be learned by SGD. Thus, our main task is to estimate the probability that we can find representatives of all target links in the ER source network, i.e., every neuron in Layer $l = 1$ has at least one edge to every block in $l = 0$ of size $q_0$, as shown in Fig. 1 (b). This probability is given by $(1 - (1 - p_1)^{q_0})^{m_{T,1}q_1}$. For the second layer, we repeat a similar argument to bound the probability $(1 - (1 - p_2)^{q_1})^{m_{T,2}}$ with $q_2 = 1$, since we do not require multiple copies of the output neurons. Bounding this probability by $1 - \delta$ completes the proof, as detailed in Appendix A.3.

Theorem 2.4 shows that $q_0$ and $q_1$ depend on $1/\log(1/\text{sparsity})$. We now generalize the idea to create multiple copies of target neurons in every layer to a fully connected network of depth $L$, which yields a similar result as above and is stated formally in Theorem 2.5.

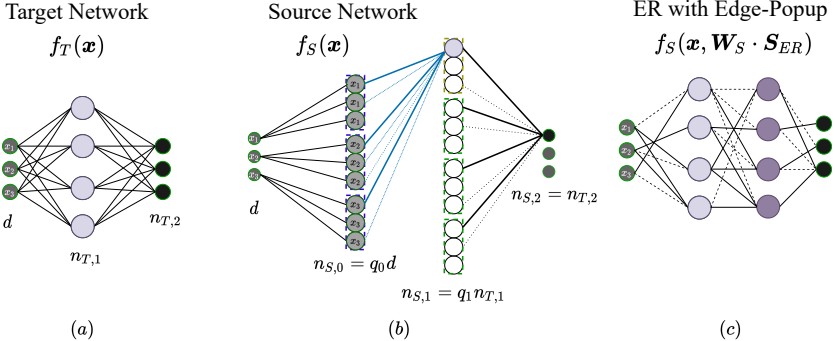

Figure 1: *LTs in ER networks:* In (a), $f_T(\boldsymbol{x})$ is a single layer target network. $(b)$ visualizes the source network $f_S(\boldsymbol{x})$, which contains a WLT. (c) shows a strong LT. The figure shows connections for only one neuron in every layer of $f_S$ for simplicity. *Dotted* and *solid* lines belong to the random mask $\boldsymbol{S}_{ER}$, while the *solid* lines belong to nonzero weights of the final LT ($\boldsymbol{S_{LT}}$).

## 2.4 Extension to $L$-layer Target Networks

Extending our insight from the 2-layer construction of the source network in the previous section, we provide a general result for a target network $f_T$ of depth $L$ and ER source networks with different layerwise sparsity ratios $p_l$. While we could approximate each target layer separately with two ER source layers, we instead present a construction that requires only one additional layer so that $L_s = L + 1$. This transfers the approach for SLTs Burkholz (2022b;a) to weak ER networks. But we have to solve two extra challenges. (a) We need to ensure that a sufficient number of neurons is

connected to the main network and can be used in the LT construction. (b) We have to show that the required number of potential matches for target neurons $q_l$ does not explode for an increasing number of layers. In fact it only scales logarithmically in the relevant variables.

**Theorem 2.5** (Existence of weak lottery tickets in ER networks ). *With Assumption 2.3, given a fully connected target network $f_T$ of depth $L$, $\delta \in (0, 1)$, a target density $p$ and a $L + 1$-layer ER source network $f_S \in ER(\mathbf{p})$ with widths $n_{S,0} = q_0 d$ and $n_{S,l} = q_l n_{T,l}, l \in \{1, 2, .., L\}$, where*

$$q_l \geq \frac{1}{\log(1/(1 - p_{l+1}))} \log\left(\frac{Lm_{T,l+1}q_{l+1}}{\delta}\right) \text{for } l \in \{0, 1, .., L - 1\} \text{ and } q_L = 1,$$

*then with probability $1 - \delta$ the random source network $f_S$ contains a weak LT $f_{WLT}$.*

*Proof Outline:* Again we follow the same procedure of finding the smallest width for every layer in the source network such that there is at least one connecting edge between a target neuron copy and one of the copies in the previous layer. Repeating this argument for every layer starting from the output layer in reverse order gives us the lower bound on the factor $q_l$ in every layer $l \in \{0, 1, .., L\}$. We can use the trainability assumption 2.3 to choose the weights of the sparse ER network such that for every target parameter there is at least one nonzero (unmasked) parameter in the source which exactly learns the required value. Full details are given in Appendix A.4.

## 2.5 EXISTENCE FOR CONVOLUTIONAL LAYERS

We can also extend our WLT existence results in ER networks to convolutional layers whose number of channels need to be overparameterized by a factor of $1/\log(1/\text{sparsity})$.

**Theorem 2.6** (Existence of WLTs for ER networks with convolutional layers). *With Assumption 2.3, given a target network $f_T$ of depth $L$ with convolutional layers $\mathbf{h}_{T,i}^{(l)} = \sum_{j=1}^{c_{l-1}} \mathbf{W}_{T,ij}^{(l)} * \mathbf{x}_{ij}^{(l-1)} + b_{T,i}^{(l)}, \mathbf{W}_T \in \mathbb{R}^{c_l \times c_{l-1} \times k_l}, \delta \in (0, 1)$, a target density $p$ and a $L + 1$-layer ER source network $f_S \in ER(\mathbf{p})$ with convolutional layers $\mathbf{h}_{S,i}^{(l)} = \sum_{j=1}^{c_{l-1}} \mathbf{W}_{S,ij}^{(l)} * \mathbf{x}_{ij}^{(l-1)} + b_{S,i}^{(l)}, \mathbf{W}_S \in \mathbb{R}^{q_l c_l \times c_{l-1} \times k_l}$ where*

$$q_l \geq \frac{1}{\log(1/(1 - p_{l+1}))} \log\left(\frac{Lm_{T,l+1}q_{l+1}}{\delta}\right) \text{for } l \in \{0, 1, .., L - 1\} \text{ and } q_L = 1,$$

*then with probability $1 - \delta$ the source network $f_S$ contains a a weak LT $f_{WLT}$.*

*Main idea*: Similarly as in case of fully-connected ER source networks, we create $q_l$ copies of every output channel of the target $c_l$ in the LT. Every filter element of the target can be learnt using the trainability assumption 2.3. Note that any tensor entry that leads to the same block is sufficient, since the convolution is a bi-linear operation so that $\sum_{i' \in I_i} \mathbf{W}_{i'j} * \mathbf{x}_i = \left(\sum_{i' \in I_i} \mathbf{W}_{i'j}\right) * \mathbf{x}_i$. Specifically, $\sum_{i' \in I_i} \mathbf{W}_{S,i'j}^{(l)}$ can represent a target element $w_{T,ije}^{(l)}$ if at least one weight $w_{S,i'je}^{(l)}$ is nonzero. Our method is visually explained in Fig. 3 alongside the full proof in Appendix A.5.

**Theoretical insights** We have shown that ER networks are provably a good starting point to find SLTs and WLTs without the overhead of multiple prune-train iterations. Our analysis furthermore reveals that ER networks are still overparametrized, as many of the random edges get pruned in the LT construction. This insight presents a theoretical justification for pruning approaches that start from random ER masks like Dynamic Sparse Training (Evci et al., 2020a; Mocanu et al., 2018).

But how far can we go with random ER masks alone? Our LT constructions suggest a need for considerable overparametrization if we wanted to start from ER masks with extreme initial sparsities $\geq 0.9$, since $1/\log(1/(1 - p_l)) \approx 1/p_l$ for $p_l << 1$. The next theorem establishes that we cannot expect to get around this $1/\log(1/(1 - p_l))$ limitation. Targeted rewiring, however, might improve even extremely sparse random masks, as we demonstrate in experiments.

**Theorem 2.7** (Lower bound on overparametrization in ER networks). *There exist univariate target networks $f_T(\mathbf{x}) = \phi(\mathbf{w}_T^T \mathbf{x} + b_T)$ that cannot be represented by a random $1$-hidden-layer ER source network $f_S \in ER(p)$ (i.e. weak or strong LT) with probability at least $1 - \delta$, if its width is $n_{S,1} < \frac{1}{\log(1/(1-p))} \log\left(\frac{1}{1-(1-\delta)^{1/d}}\right)$.*

## 3 EXPERIMENTS FOR WEAK LOTTERY TICKETS

Our existence proofs for WLTs rely on the property that ER networks are trainable in practice. We perform experiments on benchmark image datasets to validate this assumption.

**Layerwise Sparsity Ratios** There are plenty of reasonable choices for the layerwise sparsity ratios and thus ER probabilities $p_l$. Our theory applies to all of them. The optimal choice for a given source network architecture depends on the target network and thus the solution to a learning problem, which is usually unknown a-priori in practice. To demonstrate that our theory holds for different approaches and to provide practical insights into standard image classification problems, we investigate the following layerwise sparsity ratios in experiments. The simplest baseline is a globally *uniform* choice $p_l = p$. (Liu et al., 2021) have compared this choice in extensive experiments with their main proposal, ERK, which assigns $p_l \propto \frac{n_{in}+n_{out}}{n_{in}n_{out}}$ to a linear and $p_l \propto \frac{c_l+c_{l-1}+k_l}{c_l c_{l-1} k_l}$ (Mocanu et al., 2017) to a convolutional layer. In addition, we propose a *pyramidal* and *balanced* approach, which are visualized in Appendix A.14 for VGG19.

*Pyramidal*: This method emulates a property of WLTs that are obtained by IMP (Frankle & Carbin, 2019), that is the layer densities decay with increasing depth of the network. For a network of depth $L$, we use $p_l = (p_1)^l, p_l \in (0,1)$ so that $\frac{\sum_{l=1}^{l=L} p_l m_l}{\sum_{l=1}^{l=L} m_l} = p$. Given the architecture, we use a polynomial equation solver Harris et al. (2020) to obtain $p_1$ for the first layer such that $p_1 \in (0,1)$.

*Balanced*: The second layerwise sparsity method aims to maintain the same number of parameters in every layer for a given target network sparsity $p$ and source network architecture. Each neuron has a similar in- and out-degree on average. Every layer has $x = \frac{p}{L} \sum_{l=1}^{l=L} m_l$ nonzero parameters. Such an ER network can be realized with $p_l = x/m_l$. In case that $x \geq m_l$, we set $p_l = 1$.

To judge the quality of LT pruning algorithms, ER randomizations of pruned tickets have also been studied as baselines, which challenge state-of-the-art pruning algorithms Su et al. (2020); Ma et al. (2021). The corresponding sparsity ratios are computationally more cumbersome to obtain and thus of reduced practical interest. We still report comparisons with randomized Snip (Lee et al., 2018), Iterative Synflow (Tanaka et al., 2020), and IMP (Frankle & Carbin, 2019).

| Sparsity | Pyramidal | Balanced | Uniform | ERK | Snip (ER) | Synflow (ER) | IMP (ER) |
|---|---|---|---|---|---|---|---|
| 0.9 | 92.9 | **93.2** | 91.3 | 92.7 | **93.2** | 91.4 | 90 |
| 0.99 | **90.4** | 89.3 | 82.7 | 87.8 | 26.3 | 86.6 | 90.2 |
| 0.995 | 87.8 | **85.9** | 73.7 | 84.5 | 10 | 84 | 79 |
| 0.999 | 10 | **68.7** | 14.2 | 59.2 | 10 | 63.8 | 10 |

Table 2: *ER networks with different layerwise sparsities on CIFAR10 with VGG16.* We compare test accuracies of our layerwise sparsity ratios *balanced* and *pyramidal* with the uniform baseline, ERK and ER networks with layerwise sparsitiy ratios obtained by IMP, Iterative Synflow and Snip (denoted by ER). Confidence intervals are reported in Appendix A.8.

To complement (Liu et al., 2021), we conduct experiments in more extreme sparsity regimes $\geq 0.9$ to test the limit up to which ER networks are a viable alternative to more advanced but computationally expensive pruning algorithms. We show empirically that even in regimes of reduced performance, rewiring edges by Dynamical Sparse Training (DST) can improve the performance substantially, which highlights the utility of random ER masks even at extreme sparsities.

**Experimental Setup** We conduct our experiments with two datasets built for image classification tasks: CIFAR10 and CIFAR100 Krizhevsky et al. (2009). Additional experiments on Tiny Imagenet (Russakovsky et al., 2015) are reported in Appendix A.12. We train two popular architectures, VGG16 Simonyan & Zisserman (2015) and ResNet18 He et al. (2016), to classify images in the CIFAR10 dataset. On the larger CIFAR100 dataset, we use VGG19 and ResNet50. Each model is trained using SGD with learning rate 0.1 and momentum 0.9 with weight decay 0.0005 and batch size 128. We use the same hyperparameters as Ma et al. (2021) and train every model for 160 epochs. We repeat all our experiments over three runs and report averages and standard 0.95-confidence intervals, which can be found in the appendix due to space constraints. Our code builds on the work

of Liu et al. (2021); Tanaka et al. (2020) and is included in the supplement. All our experiments were run with 4 Nvidia A100 GPUs.

**Results on CIFAR10 and CIFAR100** Experiments on the CIFAR10 dataset are shown in Table 2 and on CIFAR100 in Table 3. The pyramidal and balanced methods are competitive and even outperform ERK in our experiments for sparsities up to $0.99$. Importantly, they also outperform layerwise sparsity ratios obtained by the expensive iterative pruning algorithms Synflow and IMP. However, for extreme sparsities $1 - p \geq 0.99$, the performance of ER networks drops significantly and even completely breaks down for methods like *ER Snip* and *pyramidal*. We conjecture that *ER Snip* and *pyramidal* are susceptible to layer collapse in the higher layers and even flow repair (see Appendix A.1) cannot dramatically increase the network's expressiveness. Additional results with ResNets on CIFAR10 and 100 are reported in the Appendix A.8.

| Sparsity | Pyramidal | Balanced | Uniform | ERK | Snip (ER) | Synflow (ER) | IMP (ER) |
|---|---|---|---|---|---|---|---|
| 0.5 | 73.6 | **73.9** | 72.8 | 73.6 | 73.8 | 72.6 | 71 |
| 0.8 | 73.7 | 73.5 | 71.4 | 72.8 | **74** | 71.6 | 68.2 |
| 0.9 | 72.7 | 72.5 | 69.1 | 71.9 | **73** | 70.8 | 63 |
| 0.99 | 60 | **65.3** | 55.8 | 63.8 | 1 | 62.6 | 1 |

Table 3: *ER networks on CIFAR100 with VGG19.* Extending the comparison in Table 2 to test accuracies on the CIFAR100 dataset. See Appendix for A.8 for confidence intervals.

**Dynamical Sparse Training** In order to improve the expressiveness of ER networks and achieve extremely sparse WLTs, ER networks can be rewired with the help of DST. Specifically, we use the algorithm RiGL (Evci et al., 2020a). First, we only rewire edges, which allows us to start from relatively sparse networks. Our results on CIFAR10 with VGG16 in Table 4 indicate that DST can improve even extremely sparse architectures. Usually, however, DST is started from ER networks with sparsity $0.5$. We show in the appendix that it can also be initialized at higher sparsity with insignificant losses in accuracy. In particular, initial balanced or pyramidal sparsity ratios seem to be able to improve the performance of RiGL. In Appendix A.13, we report additional experiments that further illustrate the utility of balanced and pyramidal sparsity ratios for typical DST experiments that prune ER networks. These results suggest that there is potential for tuning ER sparsity ratios to reduce computational costs and increase predictive performance.

| ER Method | Sparsity 0.99 | | Sparsity 0.995 | | Sparsity 0.999 | |
|---|---|---|---|---|---|---|
| | Original | Rewired | Original | Rewired | Original | Rewired |
| ERK | 87.8 | 90.8 | 84.5 | 88.3 | 59.2 | 74.1 |
| Balanced | 89.3 | 91.4 | 85.9 | 89.3 | **68.7** | **78.9** |
| Pyramidal | **90.4** | **92** | **87.8** | **90.6** | 10 | 9.8 |

Table 4: *ER networks rewired with DST:* Test Accuracies for an $ER(\mathbf{p})$ VGG16 network initialized with sparsity = $1 - p$ (original) and after rewiring edges with RiGL (Evci et al., 2020a; Liu et al., 2021) (rewired) on CIFAR10. Confidence intervals are reported in Appendix A.13.

## 4   CONCLUSIONS

We have systematically explained the effectiveness of random pruning and thus provided a theoretical justification for the use of Erdös-Rényi (ER) masks as strong baselines for lottery ticket pruning and starting point of dynamical sparse training (DST). This effectiveness has been demonstrated so far only experimentally for weak lottery tickets (WLT). We have proven theoretically and experimentally that ER networks also contain strong lottery tickets. This finding is also of practical interest, as initial sparse random sparse masks can avoid the computationally expensive process of pruning a dense network from scratch. Remarkably, as we could assume that ER networks are well trainable in practice, we could also prove the existence of WLTs. Our theory holds for a wide range of sparsity ratios, as we have also demonstrated in experiments. Yet, it suggests the necessity of high overparametrization in regimes of extreme sparsity. These limitations could partially be remedied by a combination of random pruning and targeted rewiring as, for instance, realized by DST.

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

# A APPENDIX

## A.1 FLOW PRESERVATION

Targeted pruning is known to be susceptible to layer collapse or just a sub-optimal use of resources (given in form of trainable parameters), when intermediary neurons receive no input despite nonzero output weights or zero output weights despite nonzero input weights. To avoid this issue, (Tanaka et al., 2020) has derived a specific data-independent pruning criterion, i.e., synaptic flow. Yet, flow preservation can also be achieved with a simple and computationally efficient random repair strategy that applies to diverse masking methods, including random ER masking.

The main idea behind this algorithm is to connect neurons (or filters) with zero in- or out-degree with at least one other randomly chosen neuron (or filter) in the network. To preserve the global sparsity, a new edge can replace a random previously chosen edge. Alternatively, ER networks with flow preservation could also be obtained by rejection sampling, which is equivalent to conditioning neurons on nonzero in- and out-degrees. To still meet the target density $p_l$, the ER probability $\tilde{p}_l$ would need to be appropriately adjusted. Our experiments reveal, however, that most randomly masked standard ResNet and VGG architectures usually perserve flows with high probability for different layerwise sparsity ratios up to sparsities $\approx 0.95$ (see Appendix A.1). The most problematic layers are the first and the last layer if the number of input channels and output neurons is relatively small. In consequence, most pruning schemes keep these layers relatively dense in general. In our theoretical derivations, we assume flow preservation in the first layer.

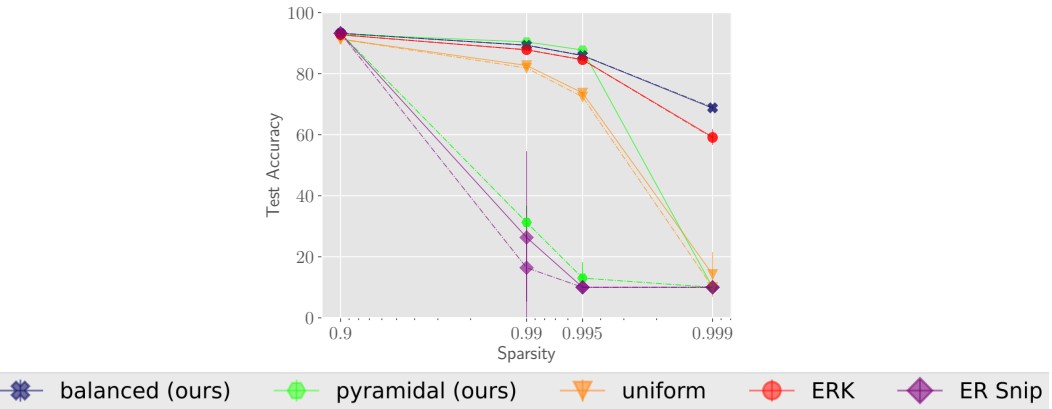

Figure 2: *Flow Comparison*: We compare the results of ER networks for each layerwise sparsity method with and without flow preservation. Solid lines denote that flow is preserved while dotted lines show the corresponding method without flow preservation for a VGG16 on CIFAR10.

We propose two methods to achieve flow preservation, which guarantees that every neuron (or filter) has at least in-degree and out-degree 1.

**Rejection Sampling**: We can resample the mask edges $s_{ij,ER}^{(l)}$ of the neurons (filters) that have a zero in-degree or a zero out-degree till there is at least one in-degree and one-out-degree for that neuron.

**Random Addition**: We randomly add an edge to a neuron with zero in-degree or out-degree. While this method adds an extra edge in the network, the total number of edges that need to be added are usually negligible in practice.

We verify the number of corrections required in an ER network to preserve flow. Notice that in most cases ER networks inherently preserve flow. For each of the used layerwise sparsity ratios, we calculate the number of connections (edges) added in the network to ensure that every neuron (or filter) has at least in-degree and one out-degree 1 using the Random Addition method. Tables 5 and 6 show the results.

| ResNet50 | Sparsity | | | | |
|---|---|---|---|---|---|
| | 0.5 | 0.8 | 0.9 | 0.99 | 0.999 |
| Uniform | 0 | 0.33 | 1 | 51.67 | 108 |
| ERK | 0 | 0 | 0 | 29.33 | 105.67 |
| ER Snip | 0 | 0 | 10.67 | 59 | 87 |
| Balanced (ours) | 0 | 0 | 0 | 23.33 | 92.33 |
| Pyramidal (ours) | 0 | 0 | 1 | 47.67 | 91 |

Table 5: Average number of mask edges added by flow correction in the ResNet18 network for CIFAR100 across three runs.

| VGG19 | Sparsity | | | | |
|---|---|---|---|---|---|
| | 0.5 | 0.8 | 0.9 | 0.99 | 0.999 |
| Uniform | 0 | 0.33 | 1.33 | 3 | 34 |
| ERK | 0 | 0 | 0 | 0 | 30.33 |
| ER Snip | 0 | 0 | 0 | 14.33 | 25 |
| Balanced (ours) | 0 | 0 | 0 | 0 | 23.33 |
| Pyramidal (ours) | 0 | 0 | 1 | 8 | 22 |

Table 6: Average number of mask edges added by flow correction in the VGG19 network for CIFAR100 averaged across three runs. Note that the number of flow corrected neurons (filters) is negligible in comparison to the number of nonzero parameters in VGG19 even for the lowest density of 0.001, which is $1, 38, 000$ parameters.

Our analysis shows that *flow preservation* is an important property that avoids layer collapse in sparse networks and is inherently satisfied in reasonable sparsity regimes $\approx 0.9$. It has a similar effect as making the final layer and the initial layer dense during pruning, which is followed in some pruning algorithms (Liu et al., 2021).

Figure 2 compares the different layerwise sparsity methods for ER networks with and without flow preservation. Our results show that flow preservation is especially important in Pyramidal and ER Snip methods. Both these methods have a higher sparsity in the final layer which leads to performance problems in case of high global sparsities. Flow preservation is able to address this partially so that a clear improvement is visible for the pyramidal method at sparsities 0.99 and 0.995.

## A.2 PROOF FOR EXISTENCE OF STRONG LOTTERY TICKETS IN ER NETWORKS

As discussed in the main manuscript, most SLT existence proofs that derive a logarithmic lower bound on the overparametrization factor of the source network utilize subset sum approximation (Lueker, 1998) in the explicit construction of a lottery ticket that approximates a target network (Pensia et al., 2020; Burkholz et al., 2022; Burkholz, 2022a; da Cunha et al., 2022; Burkholz, 2022b). We can transfer all of these proofs to ER source networks by modifying the subset sum approximation results to random variables that are set to zero with a Bernoulli probability $p$ to account for randomly missing links in the source network. We just have to replace Lueker's subset sum approximation result by Lemma 2.1 in the corresponding proofs. For simplicity, we have formulated it for uniform random variables and target parameters $z \in [-1, 1]$ but it could be easily extended to random variables that contain a uniform distribution (like normal distributions) and generally bounded targets as in Corollary 7 in (Burkholz et al., 2022). For convenience, we restate Lemma 2.1 from the main manuscript:

**Lemma A.1** (Subset sum approximation in ER Networks). *Let $X_1, ..., X_n$ be independent, uniformly distributed random variables so that $X_i \sim U([-1, 1])$ and $M_1, ..., M_n$ be independent, Bernoulli*

*distributed random variables so that $M_i \sim Ber(p)$ for a $p > 0$. Let $\epsilon, \delta \in (0, 1)$ be given. Then for any $z \in [-1, 1]$ there exists a subset $I \subset [n]$ so that with probability at least $1 - \delta$ we have $|z - \sum_{i \in I} M_i X_i| \le \epsilon$ if*

$$n \ge C \frac{1}{\log(1/(1-p))} \log\left(\frac{1}{\min(\delta, \epsilon)}\right). \tag{2}$$

**Proof** Random variables $\tilde{X}_i = M_i X_i$ do not contribute to the approximation of a target value $z$, if they are zero and thus in particular in the case that $M_i = 0$, which happens with probability $1 - p$ for each index $i$. We can thus remove all the variables $\tilde{X}_i$, for which $M_i = 0$. After a change of indexing, we arrive at a subset $\tilde{X}_1, ..., \tilde{X}_K$ of $K$ random variables, which are uniformly distributed as $\tilde{X}_i = M_i X_i = X_i \sim U([-1, 1])$, since $M_i$ is independent of $X_i$. The number of variables $K$ follows a binomial distribution, $K \sim \text{Bin}(n, p)$, since $M_1, ..., M_n$ are independent Bernoulli distributed.

For fixed $K = k$, Lueker (1998) has proven that there exists constants $a_k > 0$ and $b_k > 0$ so that the probability that the approximation is not possible is of the form $\mathbb{P}\left((\forall I \subset [k]) \, |z - \sum_{i \in I} \tilde{X}_i| > \epsilon'\right) \le a_k \exp(-b_k k)/\epsilon'$.

Using this result and defining $a := \max_{k \in [n]} a_k > 0$ and $b := \min_{k \in [n]} b_k > 0$, we just have to take an average with respect to the random variable $K \sim \text{Bin}(B, p)$.

$$\mathbb{P}\left((\forall I \subset [n]) \, |z - \sum_{i \in I} \tilde{X}_i| > \epsilon'\right) \le \sum_{k=0}^{n} \frac{a_k}{\epsilon'} \exp(-b_k k) \binom{n}{k} p^k (1-p)^{n-k}$$

$$\le \frac{a}{\epsilon'} \sum_{k=0}^{n} \binom{n}{k} \exp(-bk) p^k (1-p)^{n-k}$$

$$= \frac{a}{\epsilon'} [1 - p(1 - \exp(-b))]^n$$

To ensure the subset sum approximation is feasible with probability of at least $1 - \delta'$ we need to fulfill

$$\frac{a}{\epsilon'} [1 - p(1 - \exp(-b))]^n \le \delta'.$$

Solving for $n$ leads to

$$n \ge \frac{1}{\log\left(\frac{1}{1 - p(1 - \exp(-n))}\right)} \log\left(\frac{a}{\delta' \epsilon'}\right).$$

This inequality is satisfied if

$$n \ge C \frac{1}{\log(1/(1-p))} \log\left(\frac{1}{\min\{\delta', \epsilon'\}}\right)$$

for a generic constant $C > 0$ that depends on $a$ and $b$.

With this modified subset sum approximation, we show next that in comparison with a complete source network, an ER network needs to be wider by a factor $\frac{1}{\log(1/(1-p))}$. To provide an example of how to transfer an SLT existence proof, we focus on the construction by Burkholz (2022b).

Note that in all our theorems we assume that flow is preserved in the first layer, as it is reasonable to apply a simple and computationally cheap flow preservation algorithm after drawing a random mask (see Appendix A.1). This algorithm just ensures that all neurons are connected to the main network and are thus useful for training a neural network.

If we do not assume that flow is preserved, some neurons in the first layer might be disconnected from all input neurons with probability $(1 - p_0)^d$. Disconnected neurons could simply be ignored in the LT construction. Their share is usually negligible but, technically, without flow preservation, we would need to ensure that $n_{S,1} \ge C(1 - p_0)^d + n_{S,1}^*$, where $n_{S,1}^*$ denotes the bound on the width that we are actually going to derive.

**Theorem A.2** (Existence of SLTs in ER Networks). *Let $\epsilon, \delta \in (0,1)$, a target network $f_T$ of depth $L$, an ER($\mathbf{p}$) source network $f_S$ of depth $L+1$ with edge probabilities $p_l$ in each layer $l$ and iid initial parameters $\boldsymbol{\theta}$ with $w_{ij}^{(l)} \sim U([-1,1]), b_i^{(l)} \sim U([-1,1])$ be given. Then with probability at least $1 - \delta$, there exists a mask $\boldsymbol{S}_{LT}$ so that each target output component $i$ is approximated as $\max_{\boldsymbol{x} \in \mathcal{D}} \|f_{T,i}(\boldsymbol{x}) - f_{S,i}(\boldsymbol{x}; \boldsymbol{W}_S \cdot \boldsymbol{S}_{LT})\| \leq \epsilon$ if*

$$n_{S,l} \geq C \frac{n_{T,l}}{\log\left(1/(1-p_{l+1})\right)} \log\left(\frac{1}{\min\{\epsilon_l, \delta/\rho\}}\right)$$

*for $l \geq 1$, where $\epsilon_l = g(\epsilon, f_T)$ is defined in Equation (3) and $\rho = \frac{CN_T^{1+\gamma}}{\log(1/(1-\min_l p_l))^{1+\gamma}} \log(1/\min\{\min_l \epsilon_l, \delta\})$ for any $\gamma \geq 0$. We also require $n_{S,0} \geq Cd\frac{1}{\log(1/(1-p_1))} \log\left(\frac{1}{\min\{\epsilon_1, \delta/\rho\}}\right)$, where $C > 0$ denotes a generic constant that is independent of $n_{T,l}$, $L$, $p_l$, $\delta$, and $\epsilon$.*

Here, $\epsilon_l = g(\epsilon)$ is defined in accordance with Lemma 5.1 in (Burkholz, 2022b):

$$\epsilon_l = g(\epsilon, f_T) = \frac{\epsilon}{n_{T,L}L}\left[(1 + B_{l-1})(1 + \frac{\epsilon}{L})\prod_{k=l+1}^{L-1}(\|W_T^{(k)}\|_\infty + \frac{\epsilon}{L})\right]^{-1}, \quad B_l := \sup_{x \in \mathcal{D}}\|\boldsymbol{x}_T^{(l)}\|_1.$$

(3)

**Proof** To prove the existence of strong lottery tickets in ER networks, we modify the proof by (Burkholz, 2022b) for complete fully-connected networks.

We first answer the question, how the fact that random weights are set irreversibly to zero, changes our construction. Fig. 1 visualizes the general schematic. The general idea is that we have to create multiple copies $\rho_l$ of each target neuron in the LT, as these will enable the approximation of target parameters by utilizing subset sum approximation as modified by Lemma 2.1.

First, as Fig. 1 visualizes, we have to argue why and how we can create univariate blocks in the first layer or in general $2L$ constructions. In this case, a target layer is approximated by two appropriately pruned layers of the source network. The first of these two source layers contains only univariate neurons that form blocks that consist of neurons of the same type, which correspond to the same input target neuron $i$. All weights that start in the same block $i$ and end in the same neuron $j$ can then be utitilzed to approximate the target parameter $w_{T,ji}$. The required univariate blocks can be easily realized by pruning if flow is preserved. The reason is that each neuron in source layer $l = 0$ has at least one in-coming edge, which can survive the pruning. Since this edge could be adjacent to any of the input neurons with the same probability, we can always find enough neurons in Layer $l = 0$ that point to any of the input neurons and this allows us to form univariate blocks of similar size $B$.

Second, we have to analyze how the construction of each following target layer is affected by randomly missing edges in the source network. Each target weight $w_{T,ij}^{(l)}$ can be approximated by $w_{T,ij}^{(l)} \approx \sum_{j' \in I} m_{S,i'j'}^{(l)} w_{S,i'j'}^{(l)}$, where the neuron $i'$ in the LT approximates the target neuron $i$ and the neuron $j'$ in the LT approximates the target neuron $j$. The subset $I$ is chosen based on a modified subset sum approximation and informs the mask of the LT. Thus, $I$ exists according to Lemma 2.1, since the initially random mask entries of the source network $m_{S,i'j'}^{(l)}$ are Bernoulli distributed with probability $p_l$.

The second issue that needs to be modified for ER networks is the analysis of the number of required subset sum approximation problems $\rho$. As explained before, the main idea of the construction is to create $\rho_l$ copies of each target neuron in target Layer $l$ in Layer $l$ of the LT. These copies serve then multiple subset sum approximations to approximate the target neurons in the next layer in a similar way as the univariate blocks of the first layer. This, however, increases the total number of subset sum approximation problems $\rho$ that need to be solved and that influence the probability with which we can solve all of them. Using a union bound, we can spend $\delta/\rho$ on every approximation with a modified $\rho$ for ER networks. Similar to (Burkholz, 2022b), we can derive a lower bound on $\rho_l$ in the subsequent layers, so that the subset sum approximation is feasible for every parameter of layer $l$

when the block size $B$ is

$$B \geq \frac{1}{\log(1/(1-p_l))} \log\left(\frac{a}{\frac{\delta'}{\rho}\epsilon'}\right)$$

so that with an appropriately chosen constant $C$ we have

$$B \geq \frac{C}{\log(1/(1-p_l))} \log\left(\frac{1}{\min\{\frac{\delta'}{\rho},\epsilon'\}}\right)$$

so that it follows in total that

$$n_{S,l} \geq C \frac{n_{T,l}}{\log(1/(1-p_{l+1}))} \log\left(\frac{1}{\min\{\epsilon_l.\delta/\rho\}}\right)$$

The remaining objective is to find a $\rho \geq \rho' = \sum_{l=1}^{L} \rho'_l$, where $\rho'$ is the factor of increased subset sum approximation problems required to approximate $L$ target layers with an ER source network and $\rho_l$ counts the number of parameters in each LT layer.

Following Burkholz (2022b)'s method to identify $\rho$, we start with the last layer. The number $\rho_L$ of subset sum approximation problems that have to be solved to approximate the last layer determines the number of neurons required in the previous layer. This in turn determines the required number of neurons in the layer before it, etc. The last layer requires to solve exactly $\rho'_L = n_{T,L} n_{T,L-1}$ subset sum problems which can be solved with sufficiently high probability if $n_{S,L-1} \geq \frac{C n_{T,L-1}}{\log(1/(1-p_L))} \log(1/\min\{\epsilon_L, \delta/\rho'\})$. As we would need maximally $\frac{C}{\log(1/(1-p_L))} \log(1/\min\{\epsilon_L, \delta/\rho'\})$ sets of the target parameters in the last layer, we can bound $\rho'_{L-1} \leq \frac{C N_{L-1}}{\log(1/(1-p_L))} \log(1/\min\{\epsilon_L, \delta/\rho'\})$. Repeating the same argument for every layer, we derive $\rho'_l \leq \frac{C N_l}{\log(1/(1-p_{l+1}))} \log(1/\min\{\epsilon_{l+1}, \delta/\rho'\})$. In total, we find that $\rho' = \sum_{l=1}^{L} \rho'_l \leq \sum_{l=1}^{L} \frac{C N_l}{\log(1/(1-p_{l+1}))} \log(1/\min\{\epsilon_{l+1}, \delta/\rho'\}) \leq \frac{C N_t}{\log(1/(1-\min_l p_l))} \log(1/\min\{\min_l \epsilon_l, \delta/\rho\})$. Here, $N_l = n_{T,l} n_{T,l-1}$ and $N_t = \sum_l N_l$. A $\rho$ that fulfills $\rho \geq \frac{C N_t}{\log(1/(1-\min_l p_l))} \log(1/\min\{\epsilon_{l+1}, \delta/\rho\})$ will be sufficient. It is easy to see that $\rho = \frac{C N_T^{1+\gamma}}{\log(1/(1-\min_l p_l))^{1+\gamma}} \log(1/\min\{\min_l \epsilon_l, \delta\})$ for any $\gamma \geq 0$ fulfills our requirement.

We have thus shown the existence of SLTs in ER networks following similar ideas as the proof of Theorem 5.2 by Burkholz (2022b). Thus, our construction would also apply to more general activation functions than ReLUs. Note that we could also follow the proof strategy of Pensia et al. (2020) to show the existence of strong lottery tickets in ER networks. The key difference between the proofs of Burkholz (2022b) and Pensia et al. (2020) is how the subset sum base is created to approximate a target parameter. Pensia et al. (2020) use two layers for every layer in the target and create a basis set to approximate every target weight while Burkholz (2022b) go one step further and create multiple subset sum approximations of every target weight to avoid the two layer construction. In both these cases, the underlying subset sum approximation can be modified as shown above for ER networks and the same proof strategy as (Burkholz, 2022b) or (Pensia et al., 2020) can be followed. Similarly, we could also extend our proofs to convolutional and residual architectures (Burkholz, 2022a).

## A.3 WLT EXISTENCE PROOF FOR SINGLE HIDDEN LAYER TARGET NETWORK

**Theorem A.3** (Existence of an ER network as a WLT for a single hidden layer target network)**.** *Let Assumption 2.3 be fulfilled and a single layer fully-connected target network $f_T(\boldsymbol{x}) = \boldsymbol{W}_T^{(2)} \phi(\boldsymbol{W}_T^{(1)} \boldsymbol{x} + \boldsymbol{b}_T^{(1)}) + \boldsymbol{b}_T^{(2)}$, $\delta \in (0,1)$, a target density $p$ and a 2 layer ER source network $f_S \in ER(\mathbf{p})$ with widths $n_{S,0} = q_0 d, n_{S,1} = q_1 n_{T,1}, n_{S,2} = q_2 n_{T,2}$ be given. If*

$$q_0 \geq \frac{1}{\log(1/(1-p_1))} \log\left(\frac{2m_{T,1}q_1}{\delta}\right), q_1 \geq \frac{1}{\log(1/(1-p_2))} \log\left(\frac{2m_{T,2}}{\delta}\right) \text{ and } q_2 = 1,$$

*then with a probability $1 - \delta$ the source network $f_S$ contains a weak LT $f_{WLT} = f_S(\boldsymbol{x}; \boldsymbol{W}_S \cdot \boldsymbol{S}_{LT})$.*

**Proof of Theorem 2.4** A two hidden layer network can approximate a single hidden layer target network as explained in Section 2.4. $(q_0, q_1, q_2)$ are the overparametrization factors in each layer in the source network which ensure that we can find the links that we need in our WLT construction. Why would we need any form of overparametrization? Different from the SLT construction, we do not need to employ multiple parameters to approximate a single parameter and thus do not use any subset sum approximation. Our trainability assumption allows us to define the parameters of the ER source network so that a target network is exactly represented (i.e. with $\epsilon = 0$). Yet, we still need to prove that we can find all required nonzero entries in our mask. To increase the probability that a target link exists, we also create multiple copies of input neurons. As in the SLT construction, we prune the neurons in first layer to univariate neurons and choose the bias large enough so that the ReLU acts essentially as an identity function. $p_0 > 0$ can thus be arbitrary, as long as flow is preserved. Note that $q_2 = 1$, as the output neurons for the source and target should be identical $n_{T,2} = n_{S,2}$. The last layer (output layer) in the target contain $n_{T,2}$ neurons and the penultimate layer $n_{T,1}$. In the source network, we create $q_1$ copies of each neuron in the second layer of the target network such that $n_{S,1} = q_1 \times n_{T,1}$. Our goal is to bound the width of Layer 1 in the ER network such that there is at least one nonzero edge in the ER network for every nonzero target weight. To lower bound $q_1$, each nonzero weight $w_{T,ij}^{(2)}$ must have at least one nonzero weight (edge) in the source network with sufficiently high probability, i.e., every neuron in the output layer $n_{S,2}$ must have a nonzero edge to every block in the previous layer $n_{S,1}$ as explained in Figure 1. The probability that at least one such edge exists for each output neuron is given as $(1 - (1 - p_2)^{q_1})^{m_{T,2}}$.

Similarly, we can compute the probability that each neuron in the second layer of the source $n_{S,1}$ has at least one nonzero edge to to each of the univariate blocks in the first layer as $(1 - (1 - p_1)^{q_0})^{m_{T,1} \times q_1}$. Since each layer construction is independent from the other, the above probabilities can be multiplied to obtain the probability that we can represent the entire target network as

$$\prod_{l=0}^{2} (1 - (1 - p_l)^{q_{l-1}})^{m_{T,l} q_l} \geq 1 - \delta$$

One way to fulfill the above inequality is to split the error between the two product terms,

$$(1 - (1 - p_1)^{q_0})^{m_{T,1} q_1} \geq (1 - \delta)^{\frac{1}{2}} \text{ and } (1 - (1 - p_2)^{q_1})^{m_{T,2} q_2} \geq (1 - \delta)^{\frac{1}{2}}$$

Both equations above are satisfied with $1 - (1 - p_2)^{q_1} \geq \left(1 - \frac{\delta}{2 m_{T,2} q_2}\right)$ and $1 - (1 - p_1)^{q_0} \geq \left(1 - \frac{\delta}{2 m_{T,1} q_1}\right)$. We can now solve for $q_i, i \in \{0, 1\}$

$$q_0 \geq \frac{1}{\log(1/(1 - p_1))} \log\left(\frac{2 m_{T,1} q_1}{\delta}\right)$$

and

$$q_1 \geq \frac{1}{\log(1/(1 - p_2))} \log\left(\frac{2 m_{T,2}}{\delta}\right), \text{ since } q_2 = 1$$

After having identified a representative link in the source ER network for each target weight, we next define the weights and biases for the source ER network leveraging the trainability assumption 2.3. Each representative link in the ER source network is assigned the weight of its corresponding target. For the first layer in the source network, which is an univariate construction of the input, the weights are defined as $w_{S,ij}^{(0)} = 1$ and the bias is large enough so that all relevant inputs pass through the ReLU activation function as if it was the identity:

$$w_{S,ij}^{(0)} = 1 \ \forall j \in \{1, 2, .., d\} \text{ and } i \in \{1, 2, .., n_{S,0}\},$$

$$b_{S,i}^{(0)} = \begin{cases} -a_1 & \text{if } a_1 \leq 0 \\ 0 & \text{if } a_1 > 0 \end{cases} \text{ for every } i \in \{1, 2, .., n_{S,0}\}.$$

Recall that $a_1$ is defined as the lower bound of each input input component $\boldsymbol{x}$. We compensate for this additional bias in the last layer. Now for the second layer, every weight $w_{T,ij}^{(1)}$ in the target network is

assigned to one of the nonzero mask entries in the ER source network that lead to the corresponding input block $j$ and output block $i$. The remaining extra weights in the source are set to zero.

$$w_{S,i'j'}^{(1)} = w_{T,ij}^{(1)}, i' \in \{q_1 i, q_1 i + 1, .., q_1 i + q_1\} \text{ and } j' \in \{q_0 j, q_0 j + 1, .., q_0 j + q_0\}$$

for one pair of $i', j'$. The remaining connections between $i'$ and block $j$ can be pruned away, i.e., masked or set to zero. The bias of the second layer can be chosen so that it compensates for the extra bias added in the univariate construction of the first layer:

$$\forall i' \in \{1, ..., n_{S,1}\} \; b_{S,i'}^{(1)} = b_{T,i}^{(1)} - w_{T,ij}^{(1)} b_{S,j'}^{(0)}.$$

## A.4 WLT EXISTENCE PROOF FOR A TARGET NETWORK OF DEPTH $L$

In this section, we generalize the idea of constructing a source ER network as presented in Section 2.3 to a fully connected target network $f_T(\boldsymbol{x})$ of depth $L$. Each layer has weight $\boldsymbol{W}_T^{(l)}$ and bias $\boldsymbol{b}_T^{(l)}$. As before, we assume that the target network is equipped with the nonlinear ReLU activation function $\phi(x)$. The weight matrix has size $\boldsymbol{W}_T^{(l)} \in \mathbb{R}^{n_{T,l-1} \times n_{T,l}}$. $n_{T,l}$ is the number of neurons in each layer and $m_{T,l} = n_{T,l-1} \times n_{T,l}$ is the number of parameters in the weight matrix. Each layer has a layerwise expected density $p_l$.

**Theorem A.4** (Existence of weak lottery tickets in ER networks ). *With Assumption 2.3, given a fully connected target network $f_T$ of depth $L$, $\delta \in (0, 1)$, a target density $p$ and a $L + 1$-layer ER source network $f_S \in ER(\mathbf{p})$ with widths $n_{S,0} = q_0 d$ and $n_{S,l} = q_l n_{T,l}, l \in \{1, 2, .., L\}$, where*

$$q_l \geq \frac{1}{\log(1/(1 - p_{l+1}))} \log\left(\frac{L m_{T,l+1} q_{l+1}}{\delta}\right) \text{for } l \in \{0, 1, .., L-1\} \text{ and } q_L = 1,$$

*then with probability $1 - \delta$ the random source network $f_S$ contains a weak LT $f_{WLT} = f_S(\boldsymbol{x}; \boldsymbol{W}_S \cdot \boldsymbol{S}_{LT})$.*

**Proof for Theorem 2.5** We now construct a source network $f_S(\boldsymbol{x})$ that contains a random subnetwork which replicates $f_T(\boldsymbol{x})$ with probability $1 - \delta$. As explained in Section 2.3, we first construct an univariate layer (with index $l = 0$) in the source network assuming flow preservation.

Next, we calculate the overparametrization factor required for every layer in the source network using the same argument as A.3 starting from the last layer and working our way backwards. The last output layer has the same number of neurons in both the source and the target, $n_{S,L} = n_{T,L}$. Hence, the required width overparametrization factor is $q_L = 1$. In every intermediary layer, we create blocks of neurons that consist of $q_l$ replicates of the same target neuron. How large should $q_l$ be? In the second to last layer, the probability that each neuron in the output layer has at least one edge to each of the $q_{L-1}$ blocks in Layer $L - 1$ is $(1 - (1 - p_L)^{q_{L-1}})^{m_{T,L} q_L}$. We can similarly compute this probability for every layer all the way to the input in the source network which ensures that there is at least one edge between a neuron in every layer and each of the $q_{l-1}$ blocks in the previous layer. The probability that Layer $l$ can be constructed is thus $(1 - (1 - p_l)^{q_{l-1}})^{m_{T,l} q_l}$. These events are independent and should hold simultaneously with probability $1 - \delta$. The following inequality formalizes our argument

$$\prod_{l=1}^{L} (1 - (1 - p_l)^{q_{l-1}})^{m_{t,l} q_l} \geq 1 - \delta$$

One way to fulfill the above equation would be to ensure that

$$(1 - (1 - p_l)^{q_{l-1}})^{m_{T,l} q_l} \geq (1 - \delta)^{1/L}$$

for each layer and thus

$$(1 - (1 - p_l)^{q_{l-1}}) \geq (1 - \delta)^{1/(m_{T,l} q_l L)}$$

This inequality is fulfilled if

$$1 - (1 - p_l)^{q_{l-1}} \geq 1 - \frac{\delta}{m_{T,l} q_l L}$$

Solving for $q_{l-1}$ leads to

$$q_{l-1} \geq \frac{\log\left(\frac{\delta}{m_{T,l}q_l L}\right)}{\log(1 - p_l)} = \frac{1}{\log(1/(1 - p_l))} \log\left(\frac{Lm_{T,l}q_l}{\delta}\right).$$

We can thus compute the required width overparametrization for every layer starting from the last one, where we know $q_L = 1$. Note, that $q_l$ depends on the logarithm of $q_{l+1}$ of the next layer, which ensures that $q_l$ does not blow up as depth increases.

After making sure that the required edges exist in the ER network to represent every target weight, we still have to derive concrete parameter choices. It follows then from the trainability assumption 2.3 that these choices (or equally good ones) could be found proving the existence of WLT in ER networks.

Similar as in the single hidden layer case, each representative link in the ER source network is assigned the weight of its corresponding target and the weights in the first univariate layer are set to 1. The biases in the univariate layer are chosen so that all the inputs pass through the ReLU activation. The biases in the next layer compensate for the additional biases in the first layer.

$$w_{S,ij}^{(0)} = 1 \ \forall j \in \{1, 2, .., d\} \text{ and } i \in \{1, 2, ..., n_{S,0}\},$$

$$b_{S,i}^{(0)} = \begin{cases} -a_1 & \text{if } a_1 \leq 0, \\ 0 & \text{if } a_1 > 0 \end{cases} \text{ for every } i \in \{1, 2, ..., n_{S,0}\}.$$

For the subsequent layers in the source network $l \in \{1, 2, .., L\}$ the weights are $w_{S,i'j'}^{(l)} = w_{T,ij}^{(l)}$, where $j'$ is randomly chosen among all the non-masked connections of $i'$ to block $j$ and $j' \in \{q_{l-1}j, q_{l-1}j + 1, .., q_{l-1}j + q_{l-1}\}$ and $i' \in \{q_l i, q_l i + 1, .., q_l i + q_l\}$. The remaining connections between block $j$ and $i'$ can be pruned away or the weight parameters set to zero. The biases are set to the corresponding target bias for layers $l \in \{2, .., L\}$

$$\forall i' \in \{q_l i, q_l i + 1, .., q_l i + q_l\} \quad b_{S,i'}^{(l)} = b_{T,i}^{(l)}$$

but the second layer $l = 1$ has an additional term to compensate for the bias in the first (univariate) layer:

$$\forall i' \in \{q_l i, q_l i + 1, .., q_l i + q_l\} \quad b_{S,i'}^{(1)} = b_{T,i}^{(1)} - w_{T,ij}^{(1)}b_{S,j}^{(0)}.$$

## A.5 Existence for Convolutional Layers

**Theorem A.5** (Existence of WLTs for ER networks with convolutional layers). *With Assumption 2.3, given a target network $f_T$ of depth $L$ with convolutional layers $h_{T,i}^{(l)} = \sum_{j=1}^{c_{l-1}} W_{T,ij}^{(l)} * x_{ij}^{(l-1)} + b_{T,i}^{(l)}$, $W_T \in \mathbb{R}^{c_l \times c_{l-1} \times k_l}$, $\delta \in (0,1)$, a target density $p$ and a $L+1$-layer ER source network $f_S \in ER(\mathbf{p})$ with convolutional layers $h_{S,i}^{(l)} = \sum_{j=1}^{c_{l-1}} W_{S,ij}^{(l)} * x_{ij}^{(l-1)} + b_{S,i}^{(l)}$, $W_S \in \mathbb{R}^{q_l c_l \times c_{l-1} \times k_l}$ where*

$$q_l \geq \frac{1}{\log(1/(1 - p_{l+1}))} \log\left(\frac{Lm_{T,l+1}q_{l+1}}{\delta}\right) \text{ for } l \in \{0, 1, .., L-1\} \text{ and } q_L = 1,$$

*then with probability $1 - \delta$ the source network $f_S$ contains a a weak LT $f_{WLT}$.*

**Proof**: The linearity of convolutions allows us to construct a target filter by combining elements that are scattered between different input channels in the ER source network as shown in Figure 3. Using the same argument as the fully connected layer case, we bound the probability that at least one of the $q_l$ channels of every filter element in a convolutional weight tensor has a non-masked entry to a channel in the next layer. As for fully-connected networks, we can create blocks of channels that correspond to replicates of the same target channel. The first layer can be pruned down to univariate convolutional filters. The probability that each layer can thus be reconstructed in the convolutional network can be bounded as:

$$(1 - (1 - p_l)^{q_{l-1}})^{m_{T,l}q_l} \geq (1 - \delta)^{1/L}$$

Note that for convolutional weights, $m_{T,l}$ is the number of nonzero parameters in $\boldsymbol{W}_T^{(l)} \in \mathcal{R}^{c_l \times c_{l-1} \times k_l}$. The following width overparametrization of the output channels in a convolutional network

$$q_{l-1} \geq \frac{\log\left(\frac{\delta}{m_{T,l} q_l L}\right)}{\log(1 - p_l)} = \frac{1}{\log(1/(1-p_l))} \log\left(\frac{L m_{T,l} q_l}{\delta}\right)$$

allows an ER network to contain a WLT with probability $1 - \delta$.

The weights in the convolutional network can now be chosen using the trainability assumption 2.3 as:

$$w_{S,i'j'k}^{(l)} = w_{T,ijk}^{(l)}, \text{ for every } i' \in \{q_l i, q_l i + 1, .., q_l i + q_l\},$$

where $j' \in \{q_{l-1}j, q_{l-1}j+1, .., q_{l-1}j+q_{l-1}\}$ is chosen randomly among all non-masked connections of $i'$ to block $j$ and the remaining connections are pruned away or set to zero. The biases are set as in the proof of Theorem 2.5.

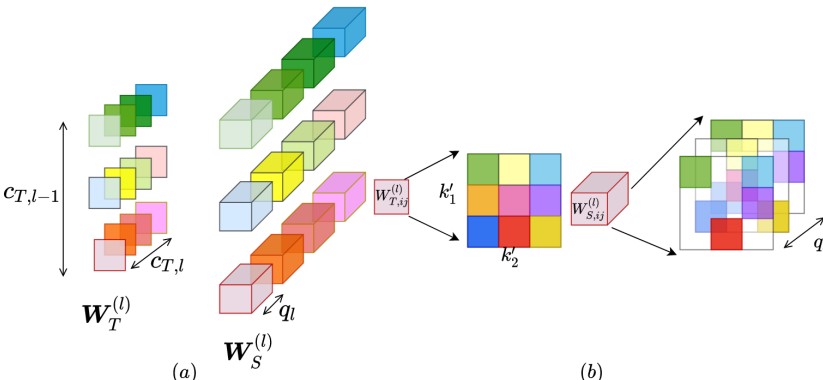

Figure 3: *Construction of a convolutional target in an ER network*: For every output channel $c_{T,l}$ in the target convolutional weight tensor $\boldsymbol{W}_T^{(l)}$, we create $q_l$ copies in the source weight tensor $\boldsymbol{W}_S^{(l)}$ as shown on the left $(a)$. The width overparametrization is further elucidated in $(b)$ where each filter element of a target output filter has $q_l$ independent copies in the source, at least one of which is nonzero (unmasked). Coloured squares in $(b)$ show the nonzero parameters in the source ER network.

## A.6 Lower Bound on the Overparametrization of ER networks

Our theoretical analysis suggests that ER networks require a width overparametrization by a factor of $\log(1/\text{sparsity})$ to exist as a LT. We also show that we cannot do substantially better than a width that is proportional to $\log(1/\text{sparsity})$.

**Theorem A.6** (Lower bound on overparametrization in ER networks). *There exist univariate target networks $f_T(\boldsymbol{x}) = \phi(\boldsymbol{w}_T^T \boldsymbol{x} + b_T)$ that cannot be represented by a random 1-hidden-layer ER source network $f_S \in ER(p)$ with probability at least $1 - \delta$, if its width is $n_{S,1} < \frac{1}{\log(1/(1-p))} \log\left(\frac{1}{1-(1-\delta)^{1/d}}\right)$.*

**Proof**: The main idea is to find the minimum width of a single hidden layer network $ER(p)$ which can approximate a single output target $f_T(\boldsymbol{x}) = \phi(\boldsymbol{w}_T^T \boldsymbol{x} + b_T)$. This minimum would be achieved when every target weight in $\boldsymbol{w}_T$ is approximated by exactly one path in the ER network from the input to the output (through the hidden layer). We derive the probability that for every weight in the target, there is at least one non-masked path in the ER source that can represent this weight as shown in Figure 4. Bounding this probability will give us a lower bound on the minimum width required in the ER network to be able to represent the target network. There are $n_{S,1}$ paths from an input neuron to an output neuron in the source network and the probability that each of this path exists is $p^2$, independently for each path, since both the input and output links in the path must be nonzero

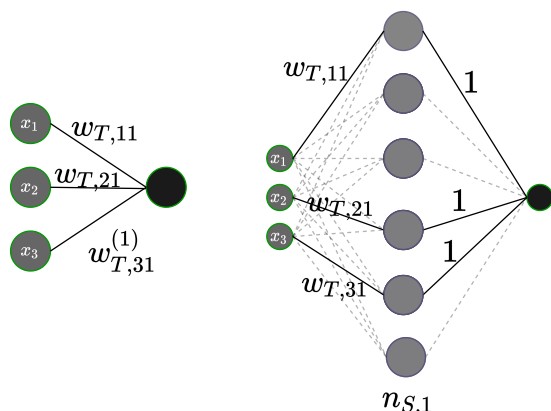

Figure 4: *Lower bound* of width of an ER source network shown on the right required to approximate the target network on the left using the trainability assumption 2.3. The solid edges in the source on the right are the nonzero (unmasked) edges while the dotted lines are masked away in an ER source network.

and each edge exists independently. Starting from the first input neuron, the probability that there is at least one path from input $x_i$ to the output is $\left(1 - (1-p^2)^{n_{S,1}}\right)$. The paths exist independently from each other if they start in different input neurons. Thus, the probability that we can represent an arbitrary target neuron with $d$ input neurons is $\left(1 - (1-p^2)^{n_{S,1}}\right)^d$. In order to find the minimum width required, we lower bound this probability as:

$$\left(1 - (1-p^2)^{n_{S,1}}\right)^d \geq 1 - \delta$$

Solving this inequality for $n_{S,1}$ proves the statement, since we would need

$$n_{S,1} \geq \frac{1}{\log(1/(1-p^2))} \log\left(\frac{1}{(1-(1-\delta)^{1/d})}\right) \geq \frac{1}{\log(1/(1-p))} \log\left(\frac{1}{(1-(1-\delta)^{1/d})}\right).$$

### A.7 EXPERIMENTAL SETUP

Our code base is built on code made available by the authors of Liu et al. (2021) and Tanaka et al. (2020).

To train ER networks as weak lottery tickets for the different layerwise sparsity ratios, we use a learning rate of $0.1$ scheduled by a factor of $1/10$ at $80$ and $120$ epochs. We train with a batch size of $128$ for $160$ epochs using SGD with momentum $0.9$ and weight decay $0.0005$. The same hyperparameter setup is used for both ResNet18 and VGG16.

For experiments on strong lottery tickets using edge popup, we use an iterative version of edge popup as described in (Fischer & Burkholz, 2022). We initialize a sparse network and anneal the sparsity iteratively while keeping the mask fixed. For the ResNet18 we use a learning rate of $0.1$ and anneal in $5$ levels and $100$ epochs for each level. The batch size is $128$ and we use SGD with momentum $0.9$ and weight decay $0.0005$. We report performances after one run for each of these experiments due to limited computation.

In the DST experiments, we use the same setup as for the ER networks stated above, and modify the mask every $100$ iterations. For sparse to sparse training with DST, we us weight magnitude as importance score for pruning (with prune rate $0.5$) and gradient for growth.

For the weak lottery ticket pruning baselines of Iterative Synflow and IMP, we prune the network in $25$ levels with $30$ epoch warm up and train the final pruned network for $160$ epochs with a learning rate $0.001$. The batch size is $256$ and the learning rate is scheduled by $0.1$ at the epochs $80$ and $120$

with the Adam(Kingma & Ba, 2015) optimizer. We use the code base of the authors of Synflow (Tanaka et al., 2020) for these experiments.

## A.8 Additional expriments on CIFAR10/100

Along with VGG we report results for different layerwise sparsity ratios for ResNets. We use a ResNet 18 for CIFAR10 and a ResNet 50 for CIFAR100.

| Sparsity | Pyramidal | Balanced | Uniform | ERK |
|---|---|---|---|---|
| 0.9 | $94.17 \pm 0.1$ | $93.97 \pm 0.13$ | $92.85 \pm 0.2$ | $93.96 \pm 0.19$ |
| 0.99 | $\mathbf{90.83 \pm 0.3}$ | $90.72 \pm 0.3$ | $84.7 \pm 0.2$ | $89.04 \pm 0.21$ |
| 0.995 | $\mathbf{88.57 \pm 0.12}$ | $88.32 \pm 0.3$ | $77.2 \pm 1$ | $85.8 \pm 0.16$ |
| 0.999 | $10 \pm 0$ | $70.69 \pm 0.6$ | $35.31 \pm 4$ | $61.36 \pm 0.24$ |

| Sparsity | Snip (ER) | Synflow (ER) | IMP (ER) |
|---|---|---|---|
| 0.9 | $\mathbf{94.25 \pm 0.3}$ | $93.95 \pm 0.13$ | $93.36 \pm 0.2$ |
| 0.99 | $90.33 \pm 0.04$ | $91.34 \pm 0.27$ | $86.43 \pm 0.4$ |
| 0.995 | $87.72 \pm 0.14$ | $88.64 \pm 0.14$ | $81.2 \pm 0.16$ |
| 0.999 | $10 \pm 0$ | $\mathbf{71.92 \pm 0.28}$ | $50.46 \pm 1.4$ |

Table 7: *ER networks with different layerwise sparsities on CIFAR10 with ResNet18.*

| Sparsity | Pyramidal | Balanced | Uniform | ERK | Snip (ER) |
|---|---|---|---|---|---|
| 0.5 | $78.09 \pm 0.64$ | $76.98 \pm 0.55$ | $\mathbf{78.12 \pm 0.33}$ | $77.63 \pm 0.52$ | $78.02 \pm 0.43$ |
| 0.8 | $\mathbf{78.44 \pm 0.41}$ | $76.59 \pm 0.33$ | $77.77 \pm 0.43$ | $77.08 \pm 0.44$ | $76.21 \pm 0.77$ |
| 0.9 | $\mathbf{76.66 \pm 0.01}$ | $75.37 \pm 0.77$ | $75.94 \pm 0.27$ | $76.02 \pm 0.62$ | $76.35 \pm 0.32$ |
| 0.99 | $65.44 \pm 1.2$ | $\mathbf{67.97 \pm 0.23}$ | $55.52 \pm 2.5$ | $65.52 \pm 0.3$ | $1 \pm 0$ |

Table 8: *ER networks with different layerwise sparsities on CIFAR100 with ResNet50.*

| Sparsity | Pyramidal | Balanced | Uniform | ERK |
|---|---|---|---|---|
| 0.9 | $92.92 \pm 0.31$ | $93.22 \pm 0.28$ | $91.31 \pm 0.3$ | $92.72 \pm 0.46$ |
| 0.99 | $\mathbf{90.41 \pm 0.03}$ | $89.31 \pm 0.1$ | $82.68 \pm 0.21$ | $87.81 \pm 0.38$ |
| 0.995 | $87.76 \pm 0.13$ | $\mathbf{85.92 \pm 0.4}$ | $73.69 \pm 0.64$ | $84.53 \pm 0.2$ |
| 0.999 | $10 \pm 0$ | $\mathbf{68.68}$ | $14.24$ | $59.22 \pm 2.6$ |

| Sparsity | Snip (ER) | Synflow (ER) | IMP (ER) |
|---|---|---|---|
| 0.9 | $\mathbf{93.23 \pm 0.2}$ | $91.4 \pm 0.11$ | $90.05 \pm 0.3$ |
| 0.99 | $26.32 \pm 28$ | $86.55 \pm 0.26$ | $90.15 \pm 0.05$ |
| 0.995 | $10 \pm 0$ | $84.03 \pm 0.06$ | $79.02 \pm 8$ |
| 0.999 | $10 \pm 0$ | $63.81 \pm 1$ | $10 \pm 0$ |

Table 9: *ER networks with different layerwise sparsities on CIFAR10 with VGG16.* We compare our layerwise sparsity ratios *balanced* and *pyramidal* with the uniform baseline, ERK and ER networks with layerwise sparsitiy ratios obtained by IMP, Iterative Synflow and Snip.

## A.9 Additional experiments for Strong Lottery Tickets in ER networks

To show experimentally that ER networks can contain SLTs, we use edge popup Ramanujan et al. (2020) to search for SLTs in ER ResNet18. We gradually anneal the sparsity of the ER network with 5 levels as proposed by (Fischer & Burkholz, 2022). The results are presented in Table 11. As a reference, we also report baseline results for dense networks in Table 12.

Additional experiments for ER VGG16 on CIFAR10 are shown in Table 13.

| Sparsity | Pyramidal | Balanced | Uniform | ERK |
|---|---|---|---|---|
| 0.5 | $73.63 \pm 0.1$ | $\mathbf{73.92 \pm 0.23}$ | $72.77 \pm 0.01$ | $73.58 \pm 0.18$ |
| 0.8 | $73.65 \pm 0.43$ | $73.51 \pm 0.25$ | $71.39 \pm 0.07$ | $72.82 \pm 0.36$ |
| 0.9 | $72.73 \pm 0.5$ | $72.49 \pm 0.43$ | $69.06 \pm 0.35$ | $71.9 \pm 0.06$ |
| 0.99 | $60.05 \pm 3.2$ | $\mathbf{65.33 \pm 0.35}$ | $55.79 \pm 0.22$ | $63.83 \pm 0.41$ |

| Sparsity | Snip (ER) | Synflow (ER) | IMP (ER) |
|---|---|---|---|
| 0.5 | $73.75 \pm 0.57$ | $72.6 \pm 0.6$ | $71.03 \pm 0.4$ |
| 0.8 | $\mathbf{74.01 \pm 0.02}$ | $71.56 \pm 0.33$ | $68.09 \pm 0.14$ |
| 0.9 | $\mathbf{73.05 \pm 0.24}$ | $70.8 \pm 0.31$ | $62.97 \pm 0.7$ |
| 0.99 | $1 \pm 0$ | $62.62 \pm 0.12$ | $1 \pm 0$ |

Table 10: *ER networks with different layerwise sparsities on CIFAR100 with VGG19.* We compare our layerwise sparsity ratios *balanced* and *pyramidal* with the uniform baseline, ERK and ER networks with layerwise sparsitiy ratios obtained by IMP, Iterative Synflow and Snip.

| Sparsity | $0.5 \rightarrow 0.8$ | $0.5 \rightarrow 0.9$ | $0.7 \rightarrow 0.9$ |
|---|---|---|---|
| Test Acc. | $87.83 \pm 0.25$ | $88.12 \pm 0.29$ | $87.95 \pm 0.25$ |

| Sparsity | $0.5 \rightarrow 0.95$ | $0.8 \rightarrow 0.95$ | $0.5 \rightarrow 0.99$ |
|---|---|---|---|
| Test Acc. | $87.78 \pm 0.33$ | $88.07 \pm 0.06$ | $87.94 \pm 0.14$ |

Table 11: *ER networks for Strong Lottery Tickets*: Average results on training an ER ResNet18 network with edge popup (Ramanujan et al., 2020) on CIFAR10. The ER network is initialized with a uniform initial sparsity, which is gradually annealed to attain a SLT of the final sparsity (initial $\rightarrow$ final sparsity). Baseline results for initially dense networks are reported in Table 12.

| Final Sparsity | $0 \rightarrow 0.8$ | $0 \rightarrow 0.9$ | $0 \rightarrow 0.95$ | $0 \rightarrow 0.99$ |
|---|---|---|---|---|
| Test Acc. | $87.79 \pm 0.1$ | $87.86 \pm 0.2$ | $88 \pm 0.3$ | $87.7 \pm 0.56$ |

Table 12: *Baseline for edge popup with ResNet18 on CIFAR10*: The results for finding a SLT using edge popup starting from a dense network are shown. Our ER results starting from a sparse network are comparable to these baseline results which validates the efficiency of ER networks.

| Sparsity | $0.5 \rightarrow 0.8$ | $0.5 \rightarrow 0.9$ | $0.5 \rightarrow 0.95$ | $0.5 \rightarrow 0.99$ |
|---|---|---|---|---|
| Test Acc. | $88.03 \pm 0.26$ | $88.31 \pm 0.29$ | $88.06 \pm 0.35$ | $88.12 \pm 0.2$ |

Table 13: *ER networks for Strong Lottery Tickets*: SLTs in VGG16 ER networks on CIFAR10. The ER network is initialized with a uniform initial sparsity and gradually annealed to attain a SLT of the final sparsity (initial $\rightarrow$ final sparsity).

| Final Sparsity | $0 \rightarrow 0.8$ | $0 \rightarrow 0.9$ | $0 \rightarrow 0.95$ | $0 \rightarrow 0.99$ |
|---|---|---|---|---|
| Test Acc. | $88.2 \pm 0.15$ | $88.38 \pm 0.14$ | $88.16 \pm 0.21$ | $88.14 \pm 0.35$ |

Table 14: *Baseline for edge popup with VGG16 on CIFAR10*: Baseline results of Edge Popup to obtain SLTs on CIFAR10 with VGG16.

### A.9.1 SLTS IN ER NETWORKS FOR RESNET110 ON CIFAR100

We find SLTs within ER networks using the Edge Popup algorithm for a larger Resnet110 model as reported in Table 15.

| ER Method | Sparsity | | |
|---|---|---|---|
| | $0 \rightarrow 0.9$ | $0.5 \rightarrow 0.9$ | $0.7 \rightarrow 0.9$ |
| Test Acc. | $61.91 \pm 0.13$ | $61.76 \pm 0.53$ | $61.78 \pm 0.61$ |

Table 15: Edge popup (SLTs) results on ER networks with Resnet110 on CIFAR100. Results are reported for one run due to limited compute.

## A.10 COMPARING ER NETWORKS WITH ITERATIVE SYNFLOW AND IMP

We have also performed experiments that compare ER networks with state of the art pruning algorithms, Iterative Synflow (Tanaka et al., 2020) and Iterative Magnitude Pruning (Frankle & Carbin, 2019). We use the iterative version of Synflow (Fischer & Burkholz, 2022) and the results are presented in Fig. 5. ER networks with layerwise sparsity ratios, which are chosen independently of the data with negligible computational overhead, are able to challenge and even outperform state-of-the-art pruning algorithms, which require computationally highly demanding pruning-training iterations. This highlights the general effectiveness of random masks at moderate sparsity levels. While random masks outperform IMP consistently in our experiments, for extreme sparsities of 0.999, Iterative Synflow still performs best among all considered algorithms. Starting Synflow from a random mask instead of a complete network, however, could potentially save computational resources to achieve similar results. We leave this analysis for future work.

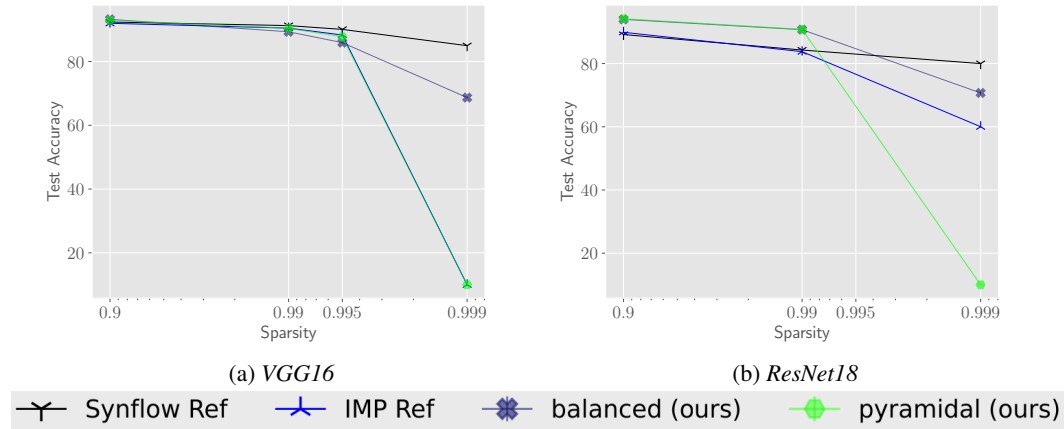

(a) *VGG16*                           (b) *ResNet18*

Figure 5: *Comparing ER networks with Iterative Synflow and IMP*: For both VGG16 and ResNet18 models the average and confidence interval over three runs is shown. The legend defines the layerwise sparsity ratio used for masking in the ER network.

## A.11 EXPERIMENTS ON CIFAR100 WITH RESNET110

To showcase the scaleability of the suggested algorithms, we additionally report experiments for a larger model, i.e., Resnet110.

### A.11.1 FOR WLTS IN ER NETWORKS

We report results for different layerwise sparsities in Table 16. As a reference we also report results on pruning with a baseline algorithm Iterative Magnitude Pruning in Table 17. We also conducted experiments with an Iterative Synflow algorithm Tanaka et al. (2020) to prune a Resnet110 but the algorithm fails for such a large model.

| ER Method | Sparsity | | | |
|---|---|---|---|---|
| | 0.5 | 0.8 | 0.9 | 0.95 |
| Balanced (ours) | $70.37 \pm 0.59$ | $67.88 \pm 1.01$ | $67.31 \pm 0.33$ | $63.80 \pm 0.09$ |
| Pyramidal (ours) | $\mathbf{71.16 \pm 0.22}$ | $69.56 \pm 0.31$ | $63.23 \pm 1.29$ | $52.37 \pm 0.51$ |
| ERK | $70.76 \pm 0.82$ | $\mathbf{69.96 \pm 0.58}$ | $\mathbf{68.14 \pm 0.34}$ | $\mathbf{64.92 \pm 0.31}$ |
| Uniform | $70.86 \pm 0.70$ | $69.41 \pm 0.27$ | $66.32 \pm 0.42$ | $61.31 \pm 0.02$ |
| ER Snip | $69.14 \pm 0.46$ | $69.12 \pm 0.65$ | $65.82 \pm 0.28$ | $60.15 \pm 0.34$ |

Table 16: *Results for WLTs in ER networks on CIFAR100 with ResNet110.* Results are average and standard deviation reported across three runs.

| Iterative Magnitude Pruning | Sparsity | |
|---|---|---|
| | 0.5 | 0.9 |
| Test Acc. | 65.46 | 64.77 |

Table 17: Results on CIFAR100 with Resnet110 pruning with the iterative magnitude pruning (IMP) algorithm for reference. Only one run of IMP was performed for each of these sparsities.

## A.12 EXPERIMENTS WITH TINY IMAGENET FOR WLTS

We also report experiments with different layerwise sparsity methods in ER networks for the Tiny Imagenet dataset. We use a VGG19 and a ResNet20 and show that our proposed layerwise sparsity methods for ER networks are competetive for this dataset. Note that we use the validation set provided by the creators of Tiny Imagenet (Russakovsky et al., 2015) as a test set to measure the generalization performance of our trained models.

See Table 18 and 19.

| ER Method | Sparsity | | | |
|---|---|---|---|---|
| | 0.5 | 0.8 | 0.9 | 0.99 |
| Balanced (ours) | $58.40 \pm 0.39$ | $57.95 \pm 0.42$ | $57.47 \pm 0.64$ | $\mathbf{50.72 \pm 0.15}$ |
| Pyramidal (ours) | $\mathbf{58.92 \pm 0.12}$ | $58.46 \pm 0.15$ | $\mathbf{58.08 \pm 0.05}$ | $41.06 \pm 0.28$ |
| ERK | $58.66 \pm 0.63$ | $58.39 \pm 0.15$ | $57.24 \pm 0.12$ | $50.52 \pm 0.50$ |
| Uniform | $58.67 \pm 0.27$ | $57.61 \pm 0.36$ | $54.96 \pm 0.82$ | $44.78 \pm 1.14$ |
| ER Snip | $58.66 \pm 0.29$ | $\mathbf{58.65 \pm 0.29}$ | $57.75 \pm 0.18$ | $0.5$ |

Table 18: *Results for ER networks on Tiny Imagenet with VGG19*

| ER Method | Sparsity | | | |
|---|---|---|---|---|
| | 0.5 | 0.8 | 0.9 | 0.99 |
| Balanced (ours) | $46.92 \pm 0.38$ | $39.25 \pm 0.57$ | $31.62 \pm 0.49$ | $9.34 \pm 0.46$ |
| Pyramidal (ours) | $48.24 \pm 0.01$ | $39.31 \pm 0.17$ | $25.15 \pm 0.08$ | $1.66 \pm 0.13$ |
| ERK | $\mathbf{50.36 \pm 0.47}$ | $\mathbf{44.73 \pm 0.64}$ | $36.81 \pm 0.32$ | $\mathbf{10.52 \pm 0.05}$ |
| Uniform | $48.49 \pm 0.47$ | $41.84 \pm 0.38$ | $32.87 \pm 0.15$ | $8.70 \pm 1.14$ |
| ER Snip | $50.28 \pm 0.50$ | $44.65 \pm 0.47$ | $\mathbf{36.92 \pm 0.20}$ | $7.36 \pm 1.81$ |

Table 19: *Results for ER networks on Tiny Imagenet with ResNet20*

## A.13 DYNAMICAL SPARSE TRAINING ON ER NETWORKS

In addition to the rewiring experiments shown in Table 4, we use Dynamical Sparse Training to prune an already sparse ER network to a higher sparsity and see if this can achieve the same performance as performing DST starting from a denser network. Similar experiments have been shown by (Liu et al., 2021.). However, we report results on ER networks starting at much higher sparsities. Our

| ER Method | Sparsity 0.99 | | Sparsity 0.995 | |
|---|---|---|---|---|
| | Original | Rewired | Original | Rewired |
| ERK | $87.81 \pm 0.39$ | $90.78 \pm 0.14$ | $84.53 \pm 0.20$ | $88.28 \pm 0.52$ |
| Balanced | $89.31 \pm 0.11$ | $91.41 \pm 0.43$ | $85.91 \pm 0.40$ | $89.30 \pm 0.03$ |
| Pyramidal | $\mathbf{90.41 \pm 0.03}$ | $\mathbf{91.97 \pm 0.08}$ | $\mathbf{87.76 \pm 0.13}$ | $\mathbf{90.61 \pm 0.15}$ |

| ER Method | Sparsity 0.999 | |
|---|---|---|
| | Original | Rewired |
| ERK | $59.22 \pm 2.57$ | $74.12 \pm 1.16$ |
| Balanced | $\mathbf{68.68 \pm 0.44}$ | $\mathbf{78.90 \pm 0.64}$ |
| Pyramidal | $10 \pm 0$ | $9.83 \pm 0.28$ |

Table 20: *ER networks rewired with DST:* An $ER(\mathbf{p})$ VGG16 network with sparsity $= 1 - p$ is initialized and the mask is modified by rewiring edges with RiGL on CIFAR10.

results shown in Table 21 are able to match the performance of (Liu et al., 2021.) while being more efficient as we start at a higher sparsity.

| ER Method | Sparsity | | |
|---|---|---|---|
| | $0.5 \rightarrow 0.99$ | $0.9 \rightarrow 0.99$ | $0.95 \rightarrow 0.99$ |
| Balanced (ours) | $93.08 \pm 0.01$ | $92.75 \pm 0.25$ | $\mathbf{92.70 \pm 0.10}$ |
| Pyramidal (ours) | $\mathbf{93.13 \pm 0.05}$ | $\mathbf{92.93 \pm 0.08}$ | $92.58 \pm 0.21$ |
| ERK | $92.94 \pm 0.12$ | $92.77 \pm 0.01$ | $92.47 \pm 0.11$ |

Table 21: *Sparse to sparse training with DST* Final test accuracy for VGG16 on CIFAR10 is reported where the model is initialized with an ER network of some initial sparsity and further pruned to a final sparsity (initial $\rightarrow$ final) while modifying the mask using the RiGL (Evci et al., 2020a) algorithm.

Notably, we observe that it is also possible to start at a sparsity of up to $0.95$ and still achieve a competitive test accuracy, only marginally worse than starting with a sparsity of $0.5$.

A.14 VISUALIZING LAYERWISE SPARSITIES FOR ER NETWORKS

We report layerwise sparsity ratios for the proposed methods discussed in Section 3 in comparison to ERK, for VGG19 with CIFAR100.

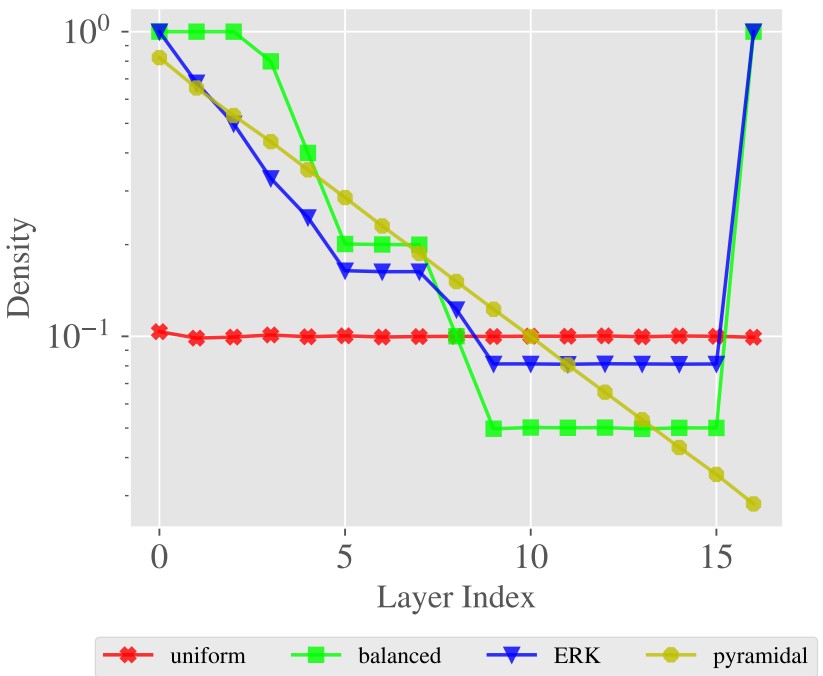

Figure 6: *Layerwise sparsity ratios for VGG19 trained on CIFAR100 for results reported in Table 3* for target sparsity $0.9$ i.e. $10\%$ of the parameters are retained.

