# OpenReview forum: "How Erdös and Rényi Win the Lottery"
_ICLR.cc/2023/Conference — Submitted to ICLR 2023_

### Official Review · Reviewer_x4uC · 2022-10-23

**Confidence:** 2
**Clarity, Quality, Novelty And Reproducibility:** Please see above.
**Correctness:** 3
**Technical Novelty And Significance:** 3
**Empirical Novelty And Significance:** 3
**Recommendation:** 6

**Strength And Weaknesses:**

**Strength**
* This paper gives solid theoretical justification on ER random network, the theory partially substantiates why some recent works on sparse networks that use ER initialization is better in practice, like (Evci et al., 2020a). I did not read all the proof, but the main claim in the paper looks correct to me.
* While I personally am not familiar with the literature in this line of work, the theory looks interesting and novel to me.

**Weakness**
* It's likely that I missed something in the literature, but it is unclear to me what the definitions for SLT and WLT are. I'd encourage the authors to clearly identify this in the revision.
* While I appreciate the authors for giving theoretical justification to ER networks, it's still unclear to me how the theory connects to the current success. Take (Evci et al., 2020a), the main success in RigL paper is that the rewiring of weight parameters is signaled by gradients instead of random variables. The authors mentioned in their contributions that theoretical support is provided on targeted rewiring, but I'm not sure how the theory given in Theorem 2.7 is related to (Evci et al., 2020a).
* The majority of the experiments are done to verify the ER trainable assumption in real-world training applications. On the other hand, I think it'd be interesting to also give evidence on how ER network contains SLT/WLT on toy examples.


**Summary Of The Paper:**

This paper provides a theoretical analysis that ER masks can approximate arbitrary target networks if they are wider by a factor of 1/log(1/s), where s denotes the sparsity ratio. The paper prove that ER randomly initialized network contains strong lottery tickets, and also prove the existing of weak lottery tickets that require a lower degree of overparameterization than strong lottery tickets. Additionally, a lower bound on overparameterization in ER networks that illustrates the limits of ER masks. The authors verify their theory on multiple DL applications like VGG on CIFAR10.

**Summary Of The Review:**

Overall I think this is a good paper, because it provides detailed theoretical justification to ER networks. I'd appreciate if the authors could provide follow-up explanations of how the provided theory relates to (Evci et al., 2020a).

---

> ### Author Response · Authors · 2022-11-09
> **Reply to Reviewer x4uC**
>
> We thank Reviewer x4uC for the positive review and are happy to elaborate on the relation between our theory and the work of [Evci et al.](https://arxiv.org/pdf/1911.11134.pdf).
>
> 1.  We start with clarifying the definitions: Weak LTs are defined as subnetworks of random source networks, which can be trained in isolation and achieve a similar performance as if we would train the dense source network. Strong LTs are also subnetworks of random source networks but perform competitively at initialization (thus they do not need to be trained any further after pruning).
> Both definitions can be found in the introduction.
>
> 2.  Liu et al. have demonstrated the `unreasonable effectiveness’ of random pruning in extensive experiments. They have shown that it can be competitive with more sophisticated but computationally expensive pruning algorithms in moderate sparsity regimes. We have proven that this is reasonable because ER masks are highly expressive and can be used to approximate any target network if wide enough.
> Pruning ER masks like Evci et al. is therefore feasible and theoretically justified. There also is a good reason why Dynamic Sparse Training usually starts from ER masks with roughly $p=0.5$ density.
> As we have shown, random ER masks are inherently limited in extreme sparsity regimes (e.g. $p \leq 0.1$ or $0.01$). Even weak LTs require overparameterized source networks and lead to overparameterized LTs. To obtain extreme sparsity levels, we further have to consider rewiring strategies of edges, for instance, as Evci et al. have proposed. These are also more effective at extreme sparsity levels, as Table 4 highlights.
>
> 3.  Further experiments on SLTs are provided in the main paper and the appendix. Please refer to Table 1 and Appendix A9.

---

### Official Review · Reviewer_jYYZ · 2022-10-25

**Confidence:** 4
**Correctness:** 3
**Technical Novelty And Significance:** 2
**Empirical Novelty And Significance:** 2
**Recommendation:** 3

**Clarity, Quality, Novelty And Reproducibility:**

**Clarity, Quality, Novelty:**
- The paper is well written and easy to understand.
- As I mentioned above, some of the claims are exaggerated in the text, so if they are modified, I think it would be a reasonable contribution.
- While the work relies heavily on past works in the Strong LTH literature, the application of the proof technique to ER graphs is novel.

**Strength And Weaknesses:**

**Strengths:**
- The premise is very interesting as they claim to prove a result about the remarkable performance of random pruning.
- As far as I am aware, a theoretical justification of the weak LTH does not exist in the literature. The authors prove a result showing this, although the assumption makes it quite unsatisfactory.
- The paper is well written and fairly easy to understand.

**Weaknesses:**
1. The paper makes it seem like they are proving a result explaining the remarkable performance of random pruning, but in reality they do not. The theorem only shows that you can further prune a random subnetwork, given sufficient overparameterization. While this is true, I don't find this particulary interesting or surprising.
    -  In the abstract as well as Section 1, they state "We offer a theoretical explanation of how such ER masks can approximate arbitrary target networks if..." - this is simply untrue. They show that within an ER mask, there exists some sort of *sub-mask* which allows you to approximate abirtrary target networks.
    -  The results of Su et al., Liu et al. and Frankle et al. show that random pruning **alone** is sufficient to reach competitive results. They do not further prune the random masks. Therefore, this theory does not imply those results whatsover.
    -  The phrasing of the results is accurate in the contributions section where they state that ER networks contain LTs with high probability. I agree with this statement although I am not sure if it is particulary interesting or surprising.
    -  The additional $\log(1/(1-p))$ factor is fairly obvious if you are familiar with the strong LTH literature. It is merely allowing for larger overparameterization to account for the missing edges from the ER mask initialization.

3. The assumption in the result about the weak LTH is far too strong. It almost assumes the theorem that they are trying to prove. I think this makes the result fairly vaccuous.
    - **Assumption 2.3** assumes that an ER network with layerwise density $p$ is trainable by SGD with standard weight initialization. This is essentially the weak LTH for random pruning. This is the result I would have liked to see proved. But it is just merely assumed. Proving the weak LTH from this seems extremely trivial.
    - They justify this assumption since it is observed in practice from works like Su et al., Ma et al. etc. However, they use it to prove a statement that supposedly explains the results from this very work. This circular reasoning seems incorrect and is akin to assuming the statement they are trying to prove.


5. I understand that the LTH literature has grown quite fast, but Section 1 is missing quite a few citations and is unfair to a few references as well.
    - The literature survey on LTH and pruning algorithms is missing Renda et al., (Comparing Rewinding and fine-tuning in Neural Network Pruning), Gale et al. (The State of Sparsity in Deep Neural Networks), Savarese et al. (Winning the Lottery with Continuous Sparsification).
    - The section on Strong LTH is again missing a few papers: Diffenderfer et al. (Multi-Prize Lottery Ticket Hypothesis), Sreenivasan et al. (Finding Nearly Everything within Random Binary Networks).
    - In Section 1, Tanaka et al. is listed under algorithms that have expensive pruning-retraining iterations but if I'm not mistaken, SynFlow uses a constant number of iterations to identify the mask at initialization. Therefore, it is nearly a single-shot pruning method.

6. There are some minor issues with claims made and terminology used in the paper.
    - It is mentioned repeatedly that strong LTs are also weak LTs. This is a somewhat imprecise statement. If you assume that weak LTs need to be trainable to high accuracy, then several strong LTs do not satisfy this requirement. Strong LTs can approximate any target network, but they do not provide any guarantees on similar trainability as the target networks. I think the literature has moved quickly and the strong LT term has caught on, but I think it should be used more carefully.
    - In Section 1.1, it is stated that the best performing WLTs are still obtained by expensive iterative pruning methods like IMP. This is not necessarily true. (See Sreenivasan et al. Rare Gems: Finding Lottery Tickets at Initialization)
    - In Section 1.1 they state that "we show that ER masks are competitive for various choices of layerwise sparsity ratios": But the results involve finding masks within ER masks. The final pruned subnetwork is not an ER graph.
    - They repeatedly claim that the theorems identify a lower bound on the width of the ER source network for which they can show existence. While this is true, it is really an upper bound on the width of the source network to guarantee existence since we have not yet proved a matching lower bound. Theorem 2.7 is indeed a lower bound.

7. Appendix A.1 Flow Preservation:
    - How often does flow preservation need to be done? Is this at every gradient step? This could get quite expensive since it needs to check every weight in the network.


**Summary Of The Paper:**

The paper proves that the Strong Lottery Ticket Hypothesis holds for FC Neural Networks and CNNs which are initialized as Erdos-Renyi (ER) graphs. It extends on the proofs by *Pensia et al.* and *Burkholz et al.* to show that logarithmic overparameterization (and an additional $\log$ factor that depends on the sparsity of the ER initialization) is sufficient to approximate any target network by pruning.
- They also claim to prove a result for the weak LTH but the assumption makes it fairly vaccuous.
- They run some experiments in the vein of Pensia et al. on CIFAR-10, CIFAR-100 and TinyImagenet. They also show that rewiring edges of a pruned network can improve the performance of the final mask significantly.

**Summary Of The Review:**

I do not think the paper in its current state should be accepted as several claims made are exaggerated and imprecise. However, if the literature survey is updated, claims modified and limitations more clearly described, I think it would be a reasonable contribution. I would be happy to change my mind if my concerns are addressed.

---

> ### Author Response · Authors · 2022-11-09
> **Reply to Reviewer jYYZ**
>
> We thank Reviewer jYYZ for the constructive feedback and the in-depth review.
>
> Before we address each point of criticism in detail, we would like to resolve a misunderstanding regarding the motivation and reasoning behind our derivations. We did not claim to explain why ER masks are well trainable. Instead, we are making statements about their **expressiveness**.
>
> Considering that the effectiveness of random pruning has been deemed unreasonable and has led to surprises in a couple of experimental papers, we are convinced that we provide a valuable contribution to the discussion, as we show that sparse random ER networks have the universal function approximation property. More precisely, we show that they can be used to approximate any target network (of stated width and depth) in a weak and a strong sense. However, random pruning is also inherently limited. We show that source networks need to be wider than a target network even if we have perfect trainability conditions. The overparameterization factor $1/log(1/(1-p))$ is required even for weak LTs unless we resort to rewinding random connections. This insight and natural limitation is currently missing from the discussion of random pruning.
>
> We have updated our manuscript to clarify this point and to avoid misunderstandings regarding the scope of our work.
>
> 1. Proof Setup for ER Networks
> - We would like to emphasize that we construct the full target network and not only its mask. In case of strong LTs, we indeed approximate the target network by pruning a random ER source network alone. In case of weak LTs, we only specify the weights and biases given an ER random mask. Some of the parameters are set to zero, which could be interpreted as further pruning. However, they can also be interpreted as the potential result of training. Why is this construction interesting despite the trainability assumption? First of all, the trainability assumption is reasonable and empirically extensively verified. Secondly, it shows that even weak LTs must be overparameterized if they are the result of random pruning.
> - Up to our knowledge, the strong LT pruning literature does not cover random masks. One of the reasons why we present results on weak LTs in addition to our proofs for strong LTs is precisely that we want to show that, in fact, the overparameterization factor $1/log(1/(1-p))$ does not originate from the `strongness’ of lottery tickets and thus the fact that we can only prune and not train the parameters. Overparameterization of ER source networks is required even under strong trainability assumptions. The familiarity of the reviewer with proofs on strong LTs should therefore not make the factor $1/log(1/(1-p))$ apparent. It does not originate from the subset sum construction. We have just integrated it in a modified subset sum construction to highlight the wide applicability of our results to different architectures and activation functions based on the established theory for strong LTs.
>
> 2.  We would like to repeat at this point that the trainability assumption leads to a relevant insight. Even weak LTs that originate in random masks must be overparameterized by a factor $1/log(1/(1-p))$. We are not trying to explain the trainability of random masks. We are investigating their expressiveness, which is also a relevant and novel contribution as explained above. We have explained this more clearly in the updated draft.
>
> 3.  We agree that the LT literature has grown quickly. We therefore have restricted our discussion to the most related works. Nevertheless, we have updated our discussion with the outlined papers and thank Reviewer jYYZ for the suggestion.
> Synflow has two versions implemented. We have used the synflow score to obtain highly performant LTs in iterative pruning/training cycles. For that reason, we have called it iterative synflow (see [Burkholz et al](https://arxiv.org/abs/2111.11153), [Frankle et al (Why are we Missing the Mark?)](https://openreview.net/pdf?id=Ig-VyQc-MLK)).

---

> > ### Author Response · Authors · 2022-11-09
> > **Reply to Reviewer jYYZ (continued)**
> >
> > 1.  We would be happy to change our terminology. However, we would need a bit more convincing, as we believe that our conceptual understanding of weak and strong LTs is correct. The reason why strong LT existence results are also considered as a contribution to the theoretical foundation of the weak LTH is that strong LTs are automatically weak LTs. Note that we assume that the target network solves a task of interest and is therefore consistent with potential training and test data. Strong LTs also solve the task of interest (as they approximate the target network). Training them would not considerably change their parameters and hence they also qualify 'after training' as solutions to our problem. (This does not imply that strong LT pruning algorithms can always achieve satisfactory performance in practice. But from a theoretical point of view, strong LTs of excellent performance exist under the derived conditions.)
> > Furthermore, note that even weak LTs need not necessarily fulfill the criterion of Reviewer jYYZ. Neither the target network nor the weak LT are required to be generally trainable in practice. In fact, it is subject of active discussion whether the parameters of LTs obtained by iterative magnitude pruning are already close to the final trained values or not (see e.g. [Frankle et al](https://arxiv.org/abs/1912.05671), [Evci et al](https://arxiv.org/abs/2010.03533)). It is not clear whether the LTs could be transferred to learning a new task. So far, mostly the transfer to similar tasks has been studied (see e.g. [Morcos et al](https://arxiv.org/abs/1906.02773)). However, this more general trainability property is usually termed `universality’.
> > 2. Otherwise, we agree and have added the citations accordingly. However, our point is more general: Any training or pruning algorithm that starts from a dense network is not as efficient as an algorithm that starts from an ER mask. We have demonstrated that the latter is feasible. For instance, this allowed us to make pruning for strong LTs more efficient. To give another example, this could also be a relevant strategy to further speed-up the search for rare gems.
> > 3. Our claims in Section 1.1 are correct. We have shown that ER masks are competitive for various choices of layerwise sparsity ratios. Note that our experiments are conducted such that we first draw a random ER mask (with different layerwise sparsity ratios) and then train the resulting masked network. From a theoretical perspective, the result of this training could also entail zero values in the parameters with a mask that is still an ER network, which why we differentiate $S_{LT}$ and $S_{ER}$ in our proofs.
> > 4. We have changed our formulations to avoid the term lower bound (except regarding Thm. 2.7), as this can lead to confusion.
> >
> >
> > **Flow Preservation**: We do not need to do flow preservation at every gradient step. We only apply our flow preservation algorithm once at initialization after the ER mask has been drawn.
> >
> > We believe that we have addressed all the comments by Reviewer jYYZ and have updated our manuscript accordingly.

---

> > > ### Comment · Reviewer_jYYZ · 2022-11-16
> > > **Response to Authors (Pt 2)**
> > >
> > > 1. Perhaps I should have been more precise. I meant to say "Trainable to high accuracy *on the given task*". This is the definition from the original LTH paper by Frankle and Carbin. But you are right, if you assume that the so-called Strong LTs already achieve the full-accuracy, then they are also LTs. My concern is more with etymology since (weak) LTs were first defined as merely Lottery Tickets. The strong lottery ticket hypothesis (Malach et al.) was proposed which then led to some of the literature renaming LTs. That said, this is a minor issue so I will concede.
> > > 3. Ah, so the trainability assumption refers to ER masks being trainable without further pruning? Then this experiment would suggest that you can simply train ER masks without any additional overparameterization? That is, suppose $f_S$ had sufficient overparameterization to find a strong LT at sparsity $p$, this would imply that I can simply initialize a random mask with sparsity ratio $p$ and it would be trainable.
> > >
> > > Thank you for the clarification.

---

> > > > ### Author Response · Authors · 2022-11-16
> > > > **Clarification of trainability assumption**
> > > >
> > > > Thank you for your continued engagement and the in-depth discussion. Of course, we are happy to clarify.
> > > > We should probably disentangle our results on strong LT, weak LTs, the trainability assumption, and when we prune.
> > > >
> > > > a) Our proofs on strong LTs are free of the trainability assumption. We show that if $f_S \in ER(p)$ and is of sufficient width, then there exists a subnetwork (that we obtain by pruning) that approximates a given target network. This network does not need any further training. It's a strong LT.
> > > >
> > > > b) In case of weak LTs, we rely on the trainability assumption that allows us to study the expressiveness of ER networks. We show that for a given random ER mask, there exist parameter values (weights and biases that could be potentially found by training) that enable the network to approximate a target network. Some of these parameters are set to zero in our specific construction, but we could also choose them to be nonzero and cancel each other out in some way. There are many possibilities and they do not necessarily correspond to pruning parts of the ER mask. But this fact does not really matter for our existence result that implies that ER masks are expressive. These results are relevant to understand the nature of the overparameterization factor $\log(1/(1-p))$. It is not really originate from pruning but results from the ER randomness and thus the fact that some of the random links won't be useful for a function representation while we need to compensate for potentially missing links.
> > > >
> > > > To summarize: Our theory on strong LTs requires no trainability assumption but relies on pruning. Our theory on weak LTs relies on the trainability assumption but not necessarily pruning. The latter is necessary to understand the limitations of ER random masks. (Even given the perfect training algorithm, we would still need the derived overparameterization factor to approximate any given target network.)

---

> > ### Comment · Reviewer_jYYZ · 2022-11-16
> > **Response to Authors**
> >
> > Thank you for your detailed response and revisions. Unfortunately, I am still not convinced about some of my primary concerns.
> >
> > 1.
> >  - Perhaps, I am misunderstanding the trainability assumption. If I assume that $f \in ER(p)$ is trainable, why can't I apply that directly to $f_S$ treating its initialization as the WLT and simply train it to approximate $f_T$? I think you are using the trainability assumption to ensure that each weight in $f_S$ can exactly be trained to match a corresponding weight of $f_T$ but that may not be necessary to approximate $f_T$ as a function. I would also recommend defining the trainability assumption more rigorously.
> > - Doesn't the $1/log(1/(1-p))$ factor follow from the modification to Leuker's result? If so, it does follow from the subset sum construction.
> >
> > 2. Once again, I believe you are proving the expressiveness of **pruning** random masks, not random masks themselves. Such a result would show that random masks alone (without any further pruning) can approximate arbitrary functions with high probability.

---

> > > ### Author Response · Authors · 2022-11-16
> > > **Reply to Response of Reviewer jYYZ**
> > >
> > > We thank the reviewer for their quick response and their willingness to engage in an in depth discussion.
> > >
> > > Reply to part 1
> > >
> > > 1. The reviewer makes a valid point, we could apply the trainability assumption to the dense source network $f_S$ with $p=1$ which would exactly learn the target network $f_T$ without requiring any overparametrization in $f_S$ under our trainability assumption. But this setting is not interesting for us, our focus is on $0<p<1$ to show that wide enough sparse networks can be trained from scratch and still approximate a target $f_T$, as experimentally observed by [Liu et al](https://arxiv.org/abs/2202.02643).
> > > 2. As rightly pointed out, the trainability assumption allows us to ensure that once trained, an ER network can learn the exact corresponding weights of the target while the remaining weights in the ER network are set to zero. While this may not be necessary to approximate $f_T$, we show that even in such an ideal training scenario, we still require the overparametrization factor of $1 / \log(1 / (1- p))$ in order for ER networks to be expressive. In doing so, we show that ER networks, while efficient and expressive, are still limited at higher sparsities.
> > > 3. The width factor  $1 / \log(1 / (1- p))$ arises from ensuring the fact that there are enough nonzero edges in a layer connected to each neuron as this factor also arises in our proof of weak lottery tickets where we do not use the subset sum approximation.
> > > 4. We are in fact showing the expressiveness of ER masks themselves, not a pruned version of them. In the case of weak lottery tickets, we show that the nonzero weights in the ER network can be learnt such that they correspond to the exact weight in the target and the remaining weights in the ER mask are set to zero, but they are still a part of the ER mask. In our experiments with ER networks too, once we initialize a network with a layerwise sparsity $p_l$, we train it to completion without further pruning, in the same fashion as Liu et al. and Ma et al.
> > >
> > > Reply to part 2
> > >
> > > 1. Yes, the trainability assumption is used to show that once initialized with sparsity $p$, an ER network is trainable (without the need for further pruning). And if the network $f_S$ is sufficiently overparametrized for the given sparsity $p$ then one of the possible ways it can approximate the target network $f_T$ after training is such that there is at least one nonzero weight in the ER network for every target weight, and the remaining weights in the ER mask are zero (detailed in Appendix A.4) i.e. the ER network is a weak lottery ticket.
> > >
> > > To summarize, our goal with the trainability assumption is to show that even in an ideal setting where a sparse ER network is trained to learn the exact weights corresponding to the target network, we would still need the source network $f_S$ to be wider by the factor $1 / \log(1 / (1- p))$ to ensure that the ER network can compute the same function as the target $f_T$, as also shown in the derivation of our lower bound (Theorem 2.7).
> > > With this assumption in place, we are able to provide relevant insights into the expressiveness of ER networks (and thus weak lottery tickets) and their limitations at extreme sparsities.
> > >
> > > We would be happy to state the trainability assumption in more detail in our manuscript to improve readability and to clearly explain how it is being used.

---

### Official Review · Reviewer_fij2 · 2022-10-27

**Confidence:** 4
**Clarity, Quality, Novelty And Reproducibility:** see my comments above.
**Correctness:** 2
**Technical Novelty And Significance:** 1
**Empirical Novelty And Significance:** 1
**Recommendation:** 1

**Strength And Weaknesses:**

+:
The problem looks interesting for purely theoretical purposes.
The theoretical results look solid though I couldn't check the proof.

-:
The practical motivation is unclear. Why would one stick with a fixed source sparse network that is randomly drawn? One can probably do much better by not having to stick with a fixed sparsity pattern.

The weak LT statement is proved with a very strong trainability assumption. Such an assumption would also imply the existence of weak LTs in the dense case too, I guess.

The theoretical contribution is not clear. Does the ER-version of subset sum result (Lemma 2.1) directly imply Theorem 2.2 using the same proof flow of Pensia et al., 2020?

Moreover, I am not sure if the ER-version of subset sum result interesting/non-trivial. Maybe I am misunderstood. Since $\frac{1}{\log{1/(1-p)}} \approx 1/p$ when $p$ is small, isn't it kinda obvious that you need $1/p$ expansion factor to solve the subset sum problem?

Or more seriously, why do we even need a separate proof for Lemma 2.1? With probability $1-\delta_1$, we can show that there will be $np(1-\epsilon_1)$ non-zero terms.  Since $M$ and $X$'s are independent, within those non-zero terms, you can solve the $\epsilon_2$-subset sum problem with probability $1-\delta_2$. By combining these two, we can easily get Lemma 2.1. Please correct me if I am wrong.

**Summary Of The Paper:**

The paper studies if LT exists in ER-like connected neural networks. Theoretical and empirical results are provided.

**Summary Of The Review:**

see my comments above.

---

> ### Author Response · Authors · 2022-11-09
> **Reply to Reviewer fij2**
>
> ### 1. Practical motivation: Why would one stick with a fixed sparsity pattern that is randomly drawn?
>
> 1. As discussed in the introduction, the success of sparse random masks is so intriguing because they offer a computationally extremely cheap pruning at initialization strategy. They work well for high enough sparsity levels so that they lead to considerable computational savings at training and test time (in comparison with a dense network but also in comparison with most other unstructured pruning schemes). They work so well that more sophisticated and computationally more expensive pruning strategies can only perform better for extreme sparsity levels (see Fig. 5 in Appendix A.10). Multiple works confirm this ([Su et al](https://arxiv.org/abs/2009.11094), [Liu et al](https://arxiv.org/abs/2202.02643)). Note that for extreme sparsity levels (>0.99), it can still be advantageous to start from random sparse masks to prune and rewire edges within a dynamic sparse training framework (see Table 4). Thus, random ER masks are still a useful starting point to obtain extremely sparse networks.
> 2. Note that the practical utility of random ER masks has so far only been realized for weak LTs. In addition, we show empirically (and prove theoretically) that also pruning for strong LTs can benefit from sparse random source networks. We can realize computational savings without hampering performance (see Table 1).
> 3. However, we have also pointed out the limitations of random pruning by providing a lower bound on the width requirement. This could, in principle, be alleviated by rewiring connections (as, e.g., by Dynamic Sparse Training, see Table 4). More on this follows in our third remark on our theoretical contributions.
>
> ### 2. Trainability Assumption
> We agree that this assumption is strong but also reasonable and empirically extensively justified. It also leads to two important insights (that are simply not relevant for dense source networks but are very relevant for sparse random ER source networks).
> 1.  Firstly, sparse random ER networks still have the universal function approximation property (as our constructive proofs demonstrate). This explains why random pruning can be successful which has been deemed unreasonable in the literature ([Liu et al](https://arxiv.org/abs/2202.02643)).
> 2.  Secondly, the random sparsity of the source network demands an increased width - even under ideal training conditions. (And we derive the required factor by which the width needs to be larger.)
>
> ### 3. Theoretical Contribution
> 1.  Lemma 2.1 could be used to transfer any existence proof for strong LTs that primarily relies on subset sum approximations, including the proof by [Pensia et al](https://arxiv.org/abs/2006.07990). We chose to formulate and prove an adapted ER subset sum result, as this points out the quite general implications of our results.
> 2.  Concerning the proof: Note that in your proposal, $\epsilon_1$ is a random variable that depends on $p$ and $\delta$. You would need to derive their exact dependence on these quantities if you would want to transfer flow of argumentation into a width requirement. Alternatively, you could also make a concentration argument and say that the width is at least $np(1-\epsilon_1)$ - but again the functional dependence on $p$ and $\delta$ matters to translate this consideration into a width requirement.
> If you want to make that flow of argumentation rigorous, you would likely end up with a similar proof that we have presented.
> 3.  Also note that we have derived the exact factor $1/log(1/(1-p))$, which can be approximated by $1/p$ only for small $p$ at **extreme** sparsity levels. This insight highlights an important limitation of ER networks at extreme sparsity levels.
> 4. In combination with our lower bound, this motivates why edge rewiring has a greater effect on very sparse random masks (see Table 4).
>
> 5.  Furthermore, note that our proof for strong LTs does not only replace the subset sum approximation by Lemma 2.1. We also have to adjust the width requirement for the increased number of subset sum approximation problems that need to be solved in the construction of [Burkholz](https://arxiv.org/pdf/2205.02321.pdf). This allows us to use a source network that has (L+1) instead of 2L in the construction of [Pensia et al](https://arxiv.org/abs/2006.07990).
>
> As the rating of Reviewer fij2 suggests that they believe that some of our results are incorrect and not novel, we kindly request to point those out. We are confident that we can solve any misunderstanding and address the criticism.

---

### Author Response · Authors · 2022-11-18
**General Response**

We thank all reviewers for their insightful comments and constructive feedback. We believe we have addressed all points of criticism and would welcome discussions and feedback. If there remain any misunderstandings or new points of criticism came up, we would be more than happy to resolve them.

We are confident that we provide novel insights into the strengths and limitations of random pruning of neural networks that are of high relevance for the ICLR community, as random pruning is a computationally extremely cheap pruning at initialization strategy. Since its ‘unreasonable effectiveness’ has puzzled the lottery ticket community in multiple experimental works, we felt that it was time to shed light on this phenomenon from a theoretical perspective.
Concretely, we show that randomly pruned ER networks are highly expressive despite their sparsity by proving the existence of strong and weak lottery tickets (LTs) in random ER networks. These indicate that, potentially, any pruning algorithm could save computational resources by starting from a randomly pruned instead of a dense network. However, all results combined have also enabled us to identify a natural limitation of random pruning in extremely sparse regimes which makes the rewiring of random edges necessary (for instance like in Dynamic Sparse Training setups). These results are based on our following innovations.

Theoretical innovations:
- We have derived an ER version of subset sum approximation results that transfers most theoretical existence results of strong LTs to random masks.
- We have provided a lower bound on the overparameterization factor that is required for the LT existence.

Experimental innovations:
- Up to our knowledge, we are the first to reduce the computational costs associated with pruning for strong LTs by pruning a sparse ER source network instead of dense one.
- We have introduced novel choices of layer-wise sparsity ratios that perform well in conjunction with random pruning.

---

### Author Response · Authors · 2022-12-08
**Gentle Reminder**

Since the discussion period is now coming to an end, we would like to thank the reviewers for their feedback and request them to follow up on our reply to the same.

We believe we have addressed all points of criticism by the reviewers and therefore kindly ask them to increase their scores accordingly.
In summary, our main contribution is to provide insights into the expressiveness of randomly pruned ER networks for both strong and weak lottery tickets. Our theoretical results explain why the effectiveness of random pruning is not ‘unreasonable’. However, we have also derived a natural limitation or random pruning in regimes of high sparsity. This limitation cannot be overcome by weak lottery ticket pruning but requires edge rewiring approaches as, for instance, implemented by Dynamic Sparse Training.

In particular, we would like to highlight as part of our results that we are the first to demonstrate empirically and prove theoretically the existence of strong lottery tickets in random ER networks. This outlines a clear way of how random pruning can reduce the required computational resources of any strong lottery ticket pruning approach.

We are therefore confident that our results are of high interest to the ICLR community.

---

### Decision · Program_Chairs · 2023-01-20

**Decision:**

Reject

**Justification For Why Not Higher Score:**

 Despite the paper being well-written, the reviewers and I are uncertain about the theoretical contribution of this work in light of its assumptions, and practical considerations.

**Justification For Why Not Lower Score:**

N/A

**Metareview: Summary, Strengths And Weaknesses:**

The paper claims that randomly generated pruning masks can create surprisingly effective sparse neural network models, and can often compete with dense architectures. These random super masks can be initially set up with small computational overhead, and can theoretically approximate any target network with a logarithmic factor in the inverse sparsity.

There are two critical reviewers, one of which has significantly engaged with the authors during the rebuttal. They note that despite the paper being well-written, it is unclear if the ER-version of subset sum result is interesting or non-trivial. Several claims are perhaps exaggerated and imprecise (the authors did work to improve it). Perhaps most importantly the assumption in the result about the Weak LTH is far too strong.

This can perhaps be a reasonable contribution for a future venue if the claims and limitations are clarified, and the text is significantly reworded to a point that is beyond this current format.